

# Influence of initial soil moisture in a Regional
# Climate Model study over West Africa. Part 2:
# Impact on the climate extremes.
Brahima KONÉ[1], Arona DIEDHIOU[1, 2], Adama Diawara[1], Sandrine Anquetin[2], N'datchoh
Evelyne Touré[1], Adama Bamba[1] and Arsene Toka Kobea[1]
[1]LAPAMF, Université Félix Houphouët Boigny, Abidjan, Côte d'Ivoire
[2]Univ. Grenoble Alpes, IRD, CNRS, Grenoble INP, IGE, F-38000 Grenoble, France
*Correspondence to:* Arona DIEDHIOU (arona.diedhiou@ird.fr)
**Abstract.**
The influence of the anomalies in initial soil moisture on the climate extreme over West Africa
is investigated using the fourth generation of Regional Climate Model coupled to the version
4.5 of the Community Land Model (RegCM4-CLM4.5). We applied the initial soil moisture on
June 1st for two summers June-July-August-September (JJAS) 2003 and JJAS 2004 (Resp. wet
and dry year in the region of interest) with 25 km of spatial resolution. We initialized the control
runs with the reanalysis soil moisture of the European Centre Meteorological Weather
Forecast's reanalysis of the 20th century (ERA20C), while for the dry and wet experiments, we
initialized the soil moisture respectively at the wilting points and field capacity. The impact on
extreme precipitation indices of the initial soil moisture, especially over the central Sahel, is
homogeneous, i.e. dry (wet) experiments tend to decrease (increase) precipitation extreme
indices only for precipitation indices related to the number of precipitation events, not for those
related to the intensity of precipitation events. Overall, the impact on temperature extremes of
the anomalies in initial soil moisture is more significant compared to precipitation extremes.
Initial soil moisture anomalies unequally affect daily minimum and maximum temperature. A
stronger impact is found on maximum temperature than minimum temperature. Over the entire
West African domain, wet (dry) experiments cause a decrease (increase) in maximum
temperature. The strongest impacts on minimum temperature indices are found mainly in wet
experiments, on the Sahara where we found the higher values of the maximum and minimum
daily minimum temperature indices (resp. TNx and TNn). The performance of RegCM4-
CLM4.5 in simulating the ten (10) extreme rainfall and temperature indices used in this study
is also highlighted.



## 1 Introduction

West Africa experienced large rainfall variability during the late 1960s. This variability leads often to flooding events, severe drought and regional heatwaves. Such extreme hydro-climatic events have major economic, environmental, and societal impacts (Easterling and al. 2000, Larsen 2003). In recent years, climate extremes have attracted much interest because they are expected to occur more frequently (International Panel on Climate Change (IPCC), 2012) than changes in mean climate. Yan and Yang (2000) show that for a large number of cases, the extreme climate changes were 5 to 10 times greater than climate mean change. Many key factors or physical mechanisms could be possible causes of the increase in climate extremes (Nicholson 1980; Le Barbé et al. 2002), such as the effect of increasing greenhouse gases in the atmosphere on the intensification of hot extremes (IPCC, 2007), the sea surface temperature (SST) anomalies (Fontaine and Janicot 1996; Folland et al., 1986), and land surface conditions (Philippon et al. 2005; Nicholson 2000). In addition, smaller-scale physical processes, including the interactions of the coupling of land-atmosphere, can also lead to changes in climate extremes. For the European summer, the influence of soil moisture in the coupling of land-atmosphere using regional climate model and focused on the extremes and trends in precipitation and temperature have been studied by Jaeger and Seneviratne (2011). For extreme temperatures, their studies have shown that interactions of soil moisture and climate have a significant impact, while for extreme precipitation, they only influence the frequency of wet days. Over Asia, Liu and al. (2014) studied the impact on subsequent precipitation and temperature of soil moisture anomalies using a regional climate model. They show that wet (dry) experiences of anomalies in initial soil moisture decrease (increase) the hot extremes, decrease (increase) the drought extremes, and increase(decrease) the cold extremes in zone of strong soil moisture-atmosphere coupling. However, none of these papers intended to examine the impacts of the anomalies in initial soil moisture on subsequent climate extreme using a regional climate model over West Africa. In the part 1, the influence of initial soil moisture on the climate mean was based on performance assessment of the Regional Climate Model coupled with the complex Community Land Model (RegCM4-CLM4.5) done by Koné and al. (2018) where the ability of the model to reproduce the climate mean has been validated. However, in the part 2, before starting to study the influence of initial soil moisture on the climate extremes, it was needed to assess first the performance of RegCM4-CLM4.5 in simulating the ten (10) temperature indices and extreme rainfall used in this study. This has never been done before over Africa. That's why we separate in two parts, to ease the reading and to come up with papers



of reasonable length. The paper is organized as follows: the section 2 describes the model
RegCM4, the experimental design and methodology used in this study; the section 3 presents
the assessment of RegCM4-CLM4.5 in climate extremes simulation and the impacts on climate
extremes of anomalies in initial soil moisture; and section 4 documents the summary and
conclusions.

## 2. Model, experimental design and methodology

### 2.1 Model description and numerical experiment

The fourth generation of the Regional Climate Model (RegCM4) of the International Centre for
Theoretical Physics (ICTP) is used in this study. Since this version, the physical representations
have been subject to a continuous process of implementation and development. The release
used in this study is RegCM4.7. The non-hydrostatic dynamical core of the MM5 (Mesoscale
Model version 5, Grell et al., 1994) has been ported to RegCM4 while maintaining the existing
hydrostatic core. We selected in this study the non-hydrostatic as model dynamical core.
RegCM4 is a limited-area model using a vertical grid sigma hydrostatic pressure coordinate
and a horizontal grid of Arakawa B-grid (Giorgi and al., 2012). The radiation scheme is from
the NCAR-CCM3 (National Center for Atmospheric Research and the Community Climate
Model Version 3) (Kiehl and al., 1996) and the aerosols representation is from Zakey and al.
(2006) and Solmon and al. (2006). The large-scale precipitation scheme used in this study is
from Pal and al. (2000), the moisture scheme is called the SUBgrid EXplicit moisture scheme
(SUBEX) which considers the sub-grid variability in clouds, the accretion and evaporation
processes for stable precipitation is from Sundqvist and al., 1989. The sensible heat and water
vapor in the planetary boundary layer over land and ocean, turbulent transports of momentum
are from Holtslag and al. (1990). The heat and moisture and the momentum of ocean surfaces
fluxes, are from Zeng and al. (1998). Convective precipitation and the land surface processes
in RegCM4.7 are represented in several options. Based on Koné and al., (2018), the convective
scheme of Emanuel (Emanuel, 1991) is used. The parameterization of the land surface
processes is from CLM4.5 (Oleson and al., 2013). In each grid cell of CLM4.5 there is 16
different plant functional types and 10 soil layers (Lawrence et al., 2011; Wang and al., 2016).
The integration of RegCM4 over the West African domain is shown in Fig. 1 with 18 vertical
levels and 25 km of horizontal resolution. The European Centre for Medium-Range Weather
Forecasts reanalysis (EIN75; Uppala and al., 2008; Simmons and al., 2007) provides the initial
and boundary conditions. The Sea Surface Temperatures (SSTs) are derived from the National



Oceanic and Atmosphere Administration optimal interpolation weekly (NOAA - OI_WK)
(Reynolds and al., 1996). The topography is derived from States Geological Survey (USGS)
Global Multi-resolution Terrain Elevation Data (GMTED; Danielson and al., 2011) at the
spatial resolution of 30 arc-second which is an update of the Global Land Cover
Characterization (GTOPO; Loveland and al., 2000) dataset.
The sensitivity of initial soil moisture is no longer than one season (Hong and Pan., 2000; Kim
and Hong, 2006). As in part I, four months (JJAS) simulation in 2003 and 2004 have been
carried out over West Africa, starting from June $1^{st}$, and the first 7 days considered as a spin-
up period (Kang and al., 2014) are excluded in the analysis. Here we focused our study on
climate extremes. The two years 2003 and 2004 have been chosen because they correspond
respectively to a wet and dry year in the region of interest and the impact of soil moisture
anomalies is investigated during the rainy season period. For each year, three experiments are
carried out, we used the soil moisture from the reanalysis of the European Centre
Meteorological Weather Forecast's Reanalysis of the $20^{th}$ century (ERA20C) to initialize the
control runs. Wet and dry experiments were initialized for the soil moisture (in volumetric
fraction $m^3.m^{-3}$) respectively at the field capacity (=0.489) and the wilting point (=$0.117.10^{-4}$)
over the West African derived from ERA20C soil moisture dataset.
**2.2 Validation datasets and evaluation metrics**
Our investigation is focused on the air temperature at 2 m and the precipitation over the West
African domain during the summer of JJAS for 2003 and JJAS 2004. The simulated
precipitation fields are validated with two observation datasets: the Climate Hazards group
Infrared Precipitation Stations (CHIRPS) dataset is from the University of California at Santa
Barbara, available from 1981 to 2020 at the 0.05° high-resolution and the Tropical Rainfall
Measuring Mission 3B43V7 (TRMM) dataset with the 0.25° high-resolution available from
1998 to 2013 (Huffman et al., 2007). We validated the 2 m temperature with two observation
datasets: the global daily temperature from the Global Telecommunication System (hereafter
GTS), gridded at the horizontal resolution of 0.5° for 1979 to 2020 (Fan Y. and Huug van den
Dool, 2008) and daily temperature from ERA-Interim (EIN) reanalysis at 0.25° of horizontal
resolution available from 1979 to 2020 (Dee et al., 2011). For the comparison of the simulations
of the model with observation datasets, we regridded all the products to 0.22° × 0.22°. We used
an interpolation of the bilinear method for this purpose (Nikulin et al., 2012).





The performance of RegCM4-CLM4.5 to simulate the extreme indices has been carried using
four selected sub-regions (Fig. 1) based on the previous work of Koné and al. (2018), they
correspond to different features of annual cycle of precipitation. We used the mean bias (MB),
which captures the small-scale differences between the simulation and the observation. The
pattern correlation coefficient (PCC) is also used as a spatial correlation between model
simulations and the observation to indicate the large-scale similarity degree.
To quantify the impact of soil moisture anomalies on climate extremes Liu and al. in their work
over Asia, used the mean biases in 5 subregions, while in our study we used the mean biases
and the probability density function (PDF, Gao et al. 2016; Jaeger and Seneviratne 2011) for
this purpose to better capture how many grid points are impacted by initial soil moisture.
The two-tailed t-test is used to investigate statistically significant differences at each grid cell
of the wet and dry sensitivity experiments with respect to the control one. The low result
obtained (10%) must only be considered as a crude estimate. Jaeger and Seneviratne (2011)
sustained that it is due to the neighboring grid points which have a spatial dependence and also
to the multiplicity problem of independent tests. We can obtain a more reliable and significant
estimation with methods of resampling (Wilks et al., 2006 and 1997). However, in our case it
is not possible to do this because of the constraints of computation and the large size of datasets
(Jaeger and Seneviratne, 2011). Therefore, we perform the land point's area-weighted fraction
with statistically significance of 10% level and we display the seasonally extreme indices maps
during the years 2003 and 2004.
**2.3. Extreme rainfall and temperature indices**
In this study, to investigate the changes in precipitation and temperature in terms of duration,
occurrence and intensity, six extreme temperature and four extreme rainfall indices are
examined using daily data of minimum and maximum temperature and daily rainfall (Table 1).
These 10 extreme indices are recommended by the Expert Team on Climate Change Detection
and Indices (ETCCDI, Peterson et al., 2001). We estimated the monthly values of the indices,
which allow investigating of the seasonal variations.

**3. Results and discussion**
**3.1. Seasonal extreme rainfall**
In this section we analyze six extreme rainfall indices based on daily precipitation in RegCM4
simulations over West Africa. All precipitation indices are calculated for JJAS 2003 and JJAS



2004. Table 2 summarizes the pattern correlation coefficient (PCC) and the mean bias (MB) of
all precipitation indices studied in this section for TRMM observation and model simulations
derived from control experiments with reanalysis initial soil moisture ERA20C with respect to
CHIRPS observation, calculated for west Sahel, central Sahel, Guinea coast and the entire West
African domain during the period JJAS 2003 and JJAS 2004.

### 3.1.1 The index of the wet days occurrence (R1mm)

Figure 2 shows the mean values of wet days occurrence (R1mm index, in days) from CHIRPS
(Fig.2a, d) and TRMM (Fig2b, e) observations and their corresponding simulated control
experiments (Fig2c, f) with the initial soil moisture derived from ERA20C reanalysis. The two
observation datasets CHIRPS (Fig. 2a, d) and TRMM (Fig.2b, e) have a similar large-scale
pattern over the West African domain with a PCC up to 0.98 (Table 2). The maximum values
of wet days occurrence are located over the regions of mountains such Cameroon mountains,
Jos plateau and the Guinea highlands, while the minimum values of R1mm index are found
over the Sahel with the number of wet days which decrease gradually from South to North.
However, although we have a similitude in their large-scale patterns, at the local scale the
magnitude and extension of these maxima and minima exhibit some differences. The TRMM
datasets underestimates the R1mm index values over the central and west Sahel, and
overestimate them over the Guinea for both JJAS 2003 and JJAS 2004 (Table 2). For instance
over the central Sahel, we observe a strong mean bias (MB) about -6.76 and 7.51 days (resp.
for JJAS 2003 and JJAS 2004, Table 2), and over the Guinea coast the MB reaches 8.89 and
10.44 days (resp. for JJAS 2003 and JJAS 2004, Table 2).
The control experiments (Fig.2c, f) reproduce well the large-scale structure of the observed
rainfall with a PCCs values reaching 0.96 and 0.95 (resp. for JJAS 2003 and JJAS 2004, Table
2) over the entire West African domain, but do exhibit some biases at the locale scale in term
of spatial extent and magnitude. The control experiment displays a large and quite
homogeneous area of maximum values of R1mm under the latitude 12°N. The control
experiments overestimate the R1mm index over most of the studied domains (Table2). The
largest mean biases are found over the Guinea coast with MB more than 53.16 and 55.46 days
(resp. for JJAS 2003 and JJAS 2004, Table 2). This overestimation of the R1mm index in
RegCM4 has been also found by Thanh and al. (2017) with RegCM4 over the Asia region.
Figure 2 (second panel) displays also changes in wet days occurrence for JJAS 2003 and JJAS
2004, for dry (Fig.2g and i, resp. for JJAS 2003 and JJAS 2004) and wet experiments (Fig.2h
and j, resp. for JJAS 2003 and JJAS 2004) compared to their control experiments associated,





193 the dotted area shows changes with statistical significance of 10% level. The dry experiments

194 (Fig.2g, i) tend to decrease the number of wet days occurrence while the wet experiments

195 (Fig.2h, j) tend to favor an increase of wet days occurrence, especially over the central Sahel

196 and a small part of west Sahel. However, over the Guinea coast sub-region, both wet and dry

197 experiments show a prevailing increase, although this increase in the dry experiments, is rather

198 weak. Indicating that the number of wet days occurrence are occurred more likely not only in

199 wet experiments but also in the dry experiments.

200 For a better quantitative evaluation, Figure 3 shows the PDF distributions of the changes in

201 R1mm index over the studied domains (shown in Fig.1), during JJAS 2003 and JJAS 2004. The

202 results essentially confirm the homogeneous impact found over the central Sahel (Fig.3a). The

203 strongest impact on the R1mm index for the dry (wet) experiments is shown over the central

204 (west) Sahel, with a decrease (an increase) of R1mm index and with a peak at -5 days (10 days)

205 for the two summers JJAS 2003 and JJAS 2004. Over the West Sahel, the Guinea coast and

206 the West African domain (resp. Fig.3b, c and d), both dry and wet experiments lead to an

207 increase. For instance over Guinea coast a peak is shown at 3 days for both wet and dry

208 experiments. The sensitivity of R1mm index to the contrast of year, showing by the lag between

209 the peaks of PDFs in wet or dry experiments, is strongest over the west Sahel (Fig.3b) reaching

210 3 days in particular in wet experiments. The wet year 2003 presents great impact as compared

211 to dry year 2004. It is worth to note that, the differences of PDF distributions over the different

212 domains studied highlight the importance to separate regions in sub-regions with homogeneous

213 precipitation for analyzing.

214 Summarizing the results of this section, a strong homogeneous impact on R1mm index is found

215 over the central Sahel, i.e. the dry experiments tend to decrease the number of wet days

216 occurrence while the wet experiments lead to increase the wet days occurrence. This result is

217 in line with previous work which sustained a strong coupling of land and atmosphere in areas

218 between wet and dry climate regimes (Zhang et al., 2011; Koster and al., 2006). However, over

219 Guinea coast, west Sahel and West African domain, both dry and wet experiments lead to cause

220 an increase. The control experiments overestimated R1mm index over all the domain studied.

221

**222 3.1.2 The simple daily intensity index (SDII)**

223 We analyze in this section the SDII index which gives the amount of precipitation mean on wet

224 days (R>1mm). Figure 4 (first panel) is the same as Fig.2 (first panel), but shows the amount

225 of precipitation mean on wet days (SDII index, in mm/day). Over the entire West African



domain, a similar large-scale pattern is observed between the two observations products
CHIRPS (Fig.4a, d) and TRMM (Fig.4b, e) with a PCC up to 0.86 for both JJAS 2003 and JJAS
2004 (Table 2).  However, the maxima spatial extension and the magnitude are not similar.
CHIRPS (Fig.4a, d) presents large values of SDII index, reaching more than 25 mm/day in the
coastline of the Gulf of Guinea, while TRMM has values not exceeding 12 mm/day over most
part of this region. On the other hand, TRMM shows large sparse values of SDII index reaching
up to 20 mm/day over the central and west Sahel, while CHIRPS has values not exceeding 12
mm/day over this region for both JJAS 2003 and JJJAS 2004. The largest biases of TRMM
with respect to CHIRPS are obtained over the Guinea coast sub-region with MB more than 13
and 14 mm/day (resp. for JJAS 2003 and JJAS 2004, Table2). The large-scale pattern of
observation products is well reproduced by the control experiments (Fig.4 c, f) with a PCC
reaching up to 0.73 and 0.77 (resp. in JJAS 2003 and JJAS 2004, Table 2) over West African
domain, despite at the locale scale, they exhibit some biases. The magnitude of SDII index is
quite underestimated not exceeding 10 mm/day over most of the domain studied, except over
the Cameroon mountains (Fig.4c, f). As a result, precipitation events are less extreme in the
control experiments. The largest mean biases are located over the Guinea coast with MB more
than -13.62 and -14.65 mm/day (resp. for JJAS 2003 and JJAS 2004, Table 2).
Figure 4 (second panel) is the same as Fig. 2 (second panel), but displays changes in mean
precipitation amount on wet days. Unlike for R1mm index, a change in the mean precipitation
amount on wet days is not homogeneous over all the studied domains. In general, a similar
alternation of increase and decrease of SDII index is shown for dry and wet experiments over
most of the domains studied (Figure 4, second panel). It is difficult at the regional level to
identify trends, however, at the local level, trends can be identified. For instance, over the
Senegal and Sierra Leone, the dry (wet) experiments tend to increase (decrease) the
precipitation amount on wet days (SDII index) for both JJAS 2003 and JJAS 2004.
As in Fig.3, Figure 5 displays PDFs of changes in SDII index. The PDFs show that a maximum
of grid points over the different domains studied not presents change in precipitation amount
on wet days for wet and dry experiments highlighted by the peak centered approximately on
zero. The SDII index is not sensitive to contrast of the year in both wet and dry experiments
over the different domains studied (Fig.5).
In summary, the control experiments underestimate the SDII index over all the domain study.
It is worth to note that precipitation events are less extreme in the control experiments (SDII





index not exceeding 10 mm/day). The impact on SDII index is not homogeneous over the entire
domain studied.

### 3.1.3 The maximum duration of dry spells (CDD).

The duration of dry spells (CDD index) which represents the number of consecutive days with
precipitation less than 1 mm/day is analyzed in this section. Figure 6 (first panel) is the same as
Fig.2 (first panel), but shows the maximum number of consecutive dry days (CDD index, in
day). CHIRPS estimates show the largest values of CDD index over the Sahara more than 50
days (Fig.6a, d), while the lowest values are located over the Guinea coast with CDD index less
than 8 days. Over the West African domain, the two fields CHIRPS and TRMM display quite
similar features over the entire West African domain with PCC more than 0.92. However, at
the local scale, the two sets of observations shown some differences. In general, these
differences concern the spatial extension especially over Sahel region. In JJAS 2003, the band
of CDD values in the range [10; 20] days is extended too far into Sahel region for TRMM than
CHIRPS. On the other hand, in JJAS 2004, TRMM (Fig.6b, e) present a narrower band of
minimum CDD index values over the Guinea coast around the latitude 10°N than CHIRPS
which extend this band over Guinea coast. TRMM observation underestimates the CDD index
over the entire West African domain, with MB about -2.29 and -1.75 days (resp. for JJAS 2003
and JJAS 2004, table2).
The control experiments (Fig.6c, f), over the entire West African domain, well reproduce the
large-scale pattern of the observed rainfall with a PCC more than 0.85 and 0.89 (resp. for JJAS
2003 and JJAS 2004, Table 1). However, in term of magnitude, some differences are shown at
the locale scale. In general, the control experiments overestimate the CDD index over the whole
West African domain, the central Sahel and west Sahel (Table2). While CDD index values are
underestimated over the Guinea Coast (Table2). For example, the control experiments
overestimate the CDD index over the West African domain with MB more than 2.63 and 7 days
(resp. for JJAS 2003 and JJAS 2004, table2).  The current parametrization of the model tends
to increase the drought extreme over the central and west Sahel and the whole West African
domain, while over the Guinea is too wet.
Figure 6 (second panel) is the same as Fig.2 (second panel), but shows changes in the maximum
lengths of consecutive dry spells (CDD index). The initial soil moisture impact on the
consecutive dry spell is homogeneous over the central and west Sahel (Fig 6, second panel), the
dry (wet) experiments tends to increase (decrease) the maximum lengths of consecutive dry
spell (CDD index). However, over Guinea coast, the dry and wet experiments lead to a
dominant decrease.
Figure 7 is the same as Fig.3, but displays the PDF distribution of the changes in CDD index.
The impact on CDD index is homogeneous over the central and west Sahel. For instance, over
the central Sahel, peaks are obtained at -6 and 2 days respectively for dry and wet experiences
(Fig.7a). The weaker and non-homogeneous impact is shown over Guinea coast and the West
African domain. For instance, over the Guinea coast, a decrease in CDD index values is found
with a peak not exceeding 2 days for both wet and dry experiments (Fig.7c). The CDD index is
sensitive to the contrast of year, especially over central Sahel and in wet experiments reaching
4 days (Fig.7a). The impact in the dry year is strong than the wet year.
In summary, RegCM4 overestimate the CDD index over most of domain studied except over
the Guinea coast. A homogeneous impact on CDD index is found over central and west Sahel,
i.e. the dry (wet) experiments increase (decrease) the maximum lengths of consecutive dry spell
(CDD index). However over the Guinea coast and West African Domain, we found a dominant
decrease of CDD index.

### 3.1.4 The maximum length of wet spells (CWD).


The persistence of wet spells (CWD index) which represents the number of consecutive days
with precipitation ≥ 1 mm/day is investigated in this section. As in Fig. 2 (first panel) but for
the maximum wet spell length (CWD index, in day), the spatial distribution of CWD index is
shown in Figure 8 (first panel). The two observed products TRMM (Fig.8b, e) and CHIRPS
(Fig.8a, d) depict a similar large-scale pattern with the PCCs reaching 0.90 and 0.87 (resp. for
JJAS 2003 and JJAS 2004, Table 2). CHIRPS observation located the maximum of CWD index
over the mountain regions such as Cameroon mountains, Jos plateau and Guinea highlands and
it is more than 20 days, while the minimum values of CWD index are found over most of the
area above the latitude 17°N and not exceed 4 days (Fig.8a, d). In general, the differences
between TRMM and CHIRPS observation concern the magnitude and the maxima extent,
which are more pronounced in TRMM than in CHIRPS. Generally, TRMM underestimate the
CWD index than CHIRPS over most of the domains studied. The largest mean bias is found
over the Guinea coast region with MB more than 2.47 and 2.38 days (resp. for JJAS 2003 and
JJAS 2004, Table 2).
The control experiments well reproduce the large-scale pattern with PCCs values reaching up
to 0.81 and 0.87 (resp. for JJAS 2003 and JJAS 2004, Table 2) over the entire West African





domain. However, at the local scale the control experiments exhibit some biases in term of
magnitude and spatial extent of these maxima and minima. Control experiments overestimate
the duration of wet days over the different domains studied. We note that this overestimation
coincides with the excessive values of R1mm index (Fig.2c, f). Therefore, the overestimation
of the model of R1mm index implies that CWD index which represents the maximum number
of consecutive days with precipitation $\geq$ 1 mm/day can only be overestimated. The strongest
mean bias is found over the Guinea coast and is more than 59.21 and 60.51 days (resp. for JJAS
2003 and JJAS 2004).
Figure 8 (second panel) is the same as Fig.2 (second panel), but displays changes in the
maximum number of consecutive wet days. As for R1mm index, over the central Sahel, the
impact is homogeneous, the dry (wet) experiments tends to decrease (increase) the maximum
lengths of consecutive wet spell (CWD index) for wet and dry years (resp JJAS 2003 and JJAS
2004). However, over Guinea and west Sahel, the changes are not homogeneous, both dry and
wet experiments lead to cause a dominant increase, in JJAS 2003 and JJAS 2004 (Fig. 8B, c).
Figure 9, as in Fig.3, but shows the PDF distribution of changes in CWD index. The results
confirm the homogeneous impact on CWD index found over the central Sahel,  the dry (wet)
experiments tends to decrease (increase) the CWD index with peaks at -10 days (15 days) for
both JJAS  2003 and JJAS 2004. However, over Guinea coast, west Sahel and West African
domain, both dry and wet experiments tend to increase the CWD index. For instance, over the
Guinea coast for wet and dry experiments peaks are respectively 12 and 2 days in JJAS 2003
and JJAS 2004. The CWD index is not sensitive in contrast of year over the different domains
studied.
Summarizing the results of this section, as in R1mm and CDD index, the CWD index is
homogeneous over the central Sahel, the dry (wet) experiments tends to decrease (increase) of
the CWD index. This result confirms the strong soil moisture impact over the transition zones
with a climate between dry and wet regimes (Zhang et al., 2011; Koster et al., 2006). Contrary
to the CDD index, over the West African Domain, west Sahel and the Guinea Coast, we found
a dominant increase of CWD index. RegCM4 overestimate the duration of wet days over all
the domains studied. This overestimation of CWD index is linked with an excessive number of
wet days as documented by Diaconescu and al. (2014).

**3.1.5 The maximum one-day precipitation accumulation (RX1day).**





The maximum one-day precipitation (RX1day) during the period JJAS 2003 and JJAS 2004 is
assessed in this section. Figure 10 (first panel) is identical to Figure 2 (first panel), but shows
the spatial distribution of the maximum 1-day precipitation index (RX1day index, in mm). The
observations datasets TRMM (Fig.10b, e) and CHIRPS (Fig.10 a, d) present a quite difference
in term of the spatial extension of the maximum values of RX1day index, although their large-
scale pattern is somewhat similar with PCC more than 0.84 for both JJAS 2003 and JJAS 2004
(Table 2). TRMM observation extends maxima of RX1day more than 80 mm over the Guinea
and the Sahel region, while CHIRPS confine them over the coastline of the Gulf of Guinea.
TRMM observation overestimates the RX1day index than CHIRPS over the entire domain
studied. The largest maximum one day precipitation is found over the central Sahel with MB
reaching 35.78 and 31.66 (resp. for JJAS 2003 and JJAS 2004, Table 2).
The control experiments (Fig.10 c, f) capture the spatial pattern with PCC values 0.50 and 0.4
(resp. JJAS 2003 and JJAS 2004, Table2).  This low coefficient of PCC has been also obtained
by Thanh and al. (2017) over Asia with RegCM4 (correlation <0.3). The models simulations
failed to capture the magnitude and the spatial extent of these maxima values of RX1day index.
The control experiments underestimate the RX1day index over all the domains studied. For the
same reason with SDII index, the RX1day index is related to the amount of precipitation, due
to the excessive light precipitation simulate by the current physical parameterization of
RegCM4, the RX1day is underestimated over the entire domain studied. The largest
underestimation is located over the Guinea coast and west Sahel. For instance, over the west
Sahel, the MB is about -38.07 and -36.67 mm (resp. JJAS  2003 and JJAS  2004, Table 2).
Figure 10 (second panel) is similar to Fig. 2 (second panel), but displays changes in maximum
one day precipitation. As for SDII index, the initial soil moisture anomalies impact on the
RX1day index is not homogeneous, a similar mixture of increase and decrease of RX1day index
is shown for dry and wet experiments over most of the domains studied (Figure 10 second
panel).
Figure 11, as in Fig.3, but shows the PDF distribution of changes in RX1day index. As in SDII
index, there is a majority of grid points which not display changes highlighted by a peak at zero
(Fig.11). The RX1day index is sensitive to the contrast of years only over the west Sahel and
in wet experiments. The impact on the precipitation amount on wet days in dry year (JJAS
2004) is more pronounced than the wet year (JJAS 2004) reaching 5 mm (Fig.11b).





In summary, for the same reason with SDII index, the RX1day index is related to the amount
of precipitation, the RX1day is underestimated over the entire domain studied. A non-
homogeneous trend is identified over the different domains studied.

**391 3.1.6 The total precipitation due to very heavy precipitation days (R95pTOT)**

We now investigated in this section, the total precipitation due to very heavy precipitation days
(R95pTOT index) during the period JJAS 2003 and JJAS 2004. Figure 12 (first panel) is the
same as in as in Fig.2 (first panel), but shows the spatial distribution of R95pTOT index. TRMM
(Fig.12b, e) and CHIRPS observations (Fig.12a, d) present a similar spatial pattern over the
entire West African domain with PCC value reaching 0.91 for both JJAS 2003 and JJAS 2004
(Table 2). However, there are some biases in their spatial extent. As for RX1day index, TRMM
observation extends maxima of R95pTOT more than 60 mm over the Guinea and the Sahel
region (Fig.10), while CHIRPS confine them over the Guinea coast. Overall, TRMM shows a
dominant overestimation than CHIRPS over the West African domain about 16.54 and 18.54
mm (resp. JJAS 2003 and JJAS 2004, Table2). The control experiments (Fig.12c, f) capture the
spatial pattern with PCC values 0.59 and 0.55 (resp. JJAS 2003 and JJAS 2004, Table2). As
with SDII and RX1day indices, the control experiments underestimate the values of the
R95pTOT index, while they overestimated the R1mm index. This is also due by the current
physical parameterization scheme of the RegCM4 model which results in a positive bias for the
number of wet days with a low precipitation threshold (e. g. 1 mm.day$^{-1}$), while for the indices
of number of wet days with a higher precipitation threshold (e. g. 10 mm.day$^{-1}$, not shown here),
it results in a negative bias. The control experiments underestimate the R95pTOT index over
the entire domain studied. The largest underestimation of R95pTOT index is located over the
Guinea coast with MB more than -43 and -46 mm (resp. for JJAS 2003 and JJAS 2004, Table2).
Figure 12 (second panel) is similar to Fig.2 (second panel), but displays changes in R95pTOT
index. The both dry and wet experiments tend to cause an increase of R95pTOT index over the
orographic regions. This means that anomalies in initial soil moisture, whether dry or wet, tend
to reinforce extreme floods.
Figure 13, as Fig.3, but shows the PDF distribution of changes in R95pTOT index. An
increasing in R95pTOT index for both wet and dry experiments is shown over most of the
domains studied. The largest change is found over the west Sahel with peak reaching 5 and 2
mm respectively for wet and dry experiments (Fig.13 b). The changes in R95pTOT index are
sensitive to the contrast of the wet and dry year reaching 2 mm (resp. JJAS 2003 and JJAS





2004), especially over west Sahel (Fig. 13a). The impact on R95pTOT index in wet year is
strong than dry year over the different domains studied.
In summary, RegCM4 underestimate the R95pTOT index over the West African domain. The
anomalies in initial soil moisture, whether dry or wet, tend to reinforce extreme floods, as
documented Liu and al. (2014) in their work over the Asia.

### 3.2. Seasonal temperature extreme indices

In this section, using daily maximum and minimum temperature, we analyze four extreme
temperature indices (Table 1) in RegCM4 simulations over West Africa. All temperature
indices are calculated for JJAS 2003 and JJAS 2004. The Table 3 summarizes the pattern
correlation coefficient (PCC) and the mean bias (MB) of all temperature indices studied in this
section for EIN reanalysis and model simulations derived from control experiments with initial
soil moisture from ERA20C reanalysis, with respect to GTS observation, calculated over the
domains presented in Fig 1, during the period JJAS 2003 and JJAS 2004.

### 3.2.1. Maximum value of daily maximum temperature (TXx index)

In this section, we analyze the maximum values of daily maximum temperature (TXx index)
for JJAS 2003 and JJAS 2004.  Figure 14 (first panel) shows the maximum value of daily
maximum temperature (TXx index in °C) from GTS observation (Fig14.a, d) and EIN
reanalysis (Fig.14b, e) for JJAS 2003 and JJAS 2004  and their corresponding simulated control
experiments (Fig.14c, f) with the initial soil moisture of the reanalysis ERA20C. The GTS
observation shows the highest values of the TXx index observed over the Sahara at more than
46° C, while the lowest values (less than 32°C) are found over the Guinea coast (Fig.14a, d).
The reanalyze of EIN have similar large-scale patterns with PCC value 0.99 over the entire
West African domain (Table 3). However, some biases are shown at the local scale in terms of
spatial extent and magnitude of these maxima and minima. The reanalysis of the EIN (Fig.14b,
e) shows lower values (less than 28°C) of the TXx index over a large area along the Guinea
coastline than the GTS estimates.  While GTS presents higher values of TXx index (up to 48°C)
and a large surface area as compared to EIN reanalysis. The reanalysis of the EIN shows a
dominant negative bias of the TXx index over most of the domains studied (Table 3).
The control experiments (Fig.14c, f) reasonably well replicate the large-scale models of the
TXx index values with PCCs up to 0.99 over the entire West African domain, but they exhibit
some bias. The control experiments are closer to the maximum and minimum values of the GTS





TXx index. The control simulations overestimate the TXx values over the central and west
Sahel and underestimate them over the Guinea coast (Table 3). For instance, the greatest
overestimation is found over the west Sahel with MB about 3.02 and 2.02°C (resp. for JJAS
2003 and JJAS 2004, Table3). However, these biases obtained for TXx index in this study are
much weak as compared to that found by Thanh and al. (2017) using RegCM4 over the Asia
which can reach approximately 8° C.
Figure 14 (second panel) displays changes in TXx index for JJAS 2003 and JJAS 2004, for dry
(Fig.14g, i, resp. for JJAS 2003 and JJAS 2004) and wet experiments (Fig.14h, j, resp. for JJAS
2003 and JJAS 2004) with respect to their corresponding control experiments, the dotted area
shows changes with statistical significance of 10% level. The impact of the anomalies in initial
soil moisture on TXx index are homogeneous over the entire West African domain, i.e. the dry
experiments lead to an increase of TXx index values while the wet experiments favor a decrease
of TXx index values. We noted that, this homogeneous impact is more pronounced in dry and
the wet experiments respectively over the Guinea coast and the central Sahel (Fig.14, second
panel).
The PDF distributions of the changes in the maximum values of daily maximum temperature
(TXx index) in JJAS 2003 and JJAS 2004, over (a) central Sahel, (b) West Sahel, (c) Guinea
and (d) West Africa derived from dry and wet experiments compared to the corresponding
control experiments are shown in Figure 15. As mentioned, the results confirm the
homogeneous impact on TXx index of the initial soil moisture anomalies over all the domains
studied. The strongest impact on TXx index of the initial soil moisture anomalies is shown over
the central Sahel (Fig.15a) with a decrease (increase), with peak at -2.5°C (more than 1°C) in
wet (dry) experiments. The inter-comparison of JJAS 2003 and JJAS 2004 show that change in
TXx index is sensitive to the contrast of year, especially in dry experiments over the central
Sahel reaching 0.8°C (Fig.15a). The impact on TXx index for the dry year (JJAS 2004) is strong
than the wet year (JJAS 2003).
In summarizing this section, a homogeneous impact on TXx index is found over the whole
West African domain, i.e. the dry (wet) experiments decrease (increase) the change in TXx
index. RegCM4 overestimate and underestimate the TXx index respectively over the Sahel
(west and central) and Guinea coast.



### 3.2.2. The Minimum value of daily maximum temperature (TXn).


In this section, we analyze the minimum values of daily maximum temperature (TXn index)
for JJAS 2003 and JJAS 2004. Figure 16 (first panel) is the same as in Fig.14 (first panel), but
presents the spatial distribution of the TXn index. GTS observation (Fig.16a, d) and EIN
reanalysis (Fig.16b, e) display similar features with PCC reaching 0.99 (for both JJAS 2003
and JJAS 2004, Table 3). The maxima and minima values of TXn index are located for both
respectively over the Sahara and the Guinea coast. However, some difference can be noticed at
the local scale in terms of spatial extent and magnitude. EIN reanalysis presents a larger spatial
extent of these maxima (greater than 36°C) and minima (less than 24°C) than GTS observation.
The reanalyze of EIN show a dominant negative bias value over Guinea coast and west Sahel
(for both JJAS 2003 and JJAS 2004 Table3). For instance, over the Guinea coast with MB about
-0.70 and -1.38°C (resp. for JJAS 2003 and JJAS 2004, Table 3).
The control experiments show a good agreement with the observed (GTS) general spatial
patterns with PCC about 0.99, however overestimate the magnitude of the TXn index over all
the domains studied. For instance, over the whole West African domain, the MB is about 5.65
and 4.14°C (resp. JJAS 2003 and JJAS 2004, Table 3). As compared to a similar study carry
out by Thanh and al. (2017) over the Asia, the biases obtained in this study are weaker.
As in Fig.14 (second panel), but for changes in TXn index, is shown in the Figure 16 (second
panel). The impact on TXn index of the initial soil moisture anomalies, as for TXx index are
homogeneous over the entire West African domain, i.e. the dry experiments lead to an increase
of TXn index values while the wet experiments favor a decrease of TXn index values. The
strongest impact on TXn index is shown in wet experiments above the latitude 15 °N, especially
for JJAS 2003.
Figure 17 is the same as Fig.15, but displays the PDF distribution of changes in TXn index. As
for TXx index, the impact on TXn index to soil moisture anomalies is homogeneous over most
of the domain studied, although this impact is rather weak as compared to the TXx index. The
strongest impact on TXn index for wet experiments are found over the wet Sahel about -2°C,
while in dry experiments, it is found over the central Sahel not exceed 1° C. In addition, the
changes in TXn index are sensitive to the contrast of year, especially in dry experiments over
west Sahel reaching 0.8°C (Fig. 13b). The impact on TXn index in dry year is strong than wet
year over west Sahel.
In summary, RegCM4 overestimate the TXn index over the whole West African domain. As
for TXx index, the impact on TXn index to soil moisture anomalies is homogeneous, i.e. the



dry (wet) experiments tend to cause an increase (decrease) of TXn index values over most of
the domain studied. We noted that the impact on TXn index of the initial soil moisture
anomalies is weak as compared with TXx index.

**3.2.3. The Minimum value of daily minimum temperature (TNn).**
In this section, we analyze the minimum values of daily maximum temperature (TNn index)
for JJAS 2003 and JJAS 2004. Figure 18 (first panel) is the same as in Fig.14 (first panel), but
displays the spatial distribution of the TNn index. GTS observation (Fig.18 a, d) shows the
maxima of TNn index values above the latitude 15° N not exceeding 27° C, while the minima
values are less than 17°C and located over the mountain regions such as Cameroon mountain,
Jos Plateau and Guinea Highland. The reanalysis of EIN shows similar spatial patterns with
GTS observation, with PCC value about 0.99 over the whole West African domain (Table 3)
despite some biases. The reanalysis of EIN (Fig.18 b, e) displays a highest value of TNn index
(exceeding 27°C) than GTS estimates and located them over large areas above the latitude 15°
N. The reanalysis of EIN also shows the lowest values (less than 21°C) of TNn index than GTS
observation located over the orographic regions. The reanalysis of EIN overestimates the TNn
index values over most of the domain studied. For instance, over the West African domain with
MB reaching 3.15 and 3.11°C (resp. for JJAS 2003 and JJAS 2004, Table 3).
The control experiments (Fig.18 c, f) show a good agreement with GTS observation with PCC
values about 0.99, but do exhibited some biases. The control experiments overestimate the
magnitude of the TNn index over all the domains studied. For instance, over the whole West
African domain, the MB is about 1.45 and 0.71°C (resp. for JJAS 2003 and JJAS 2004, Table
3). These dominant positive biases obtained in simulating the TXx, TXn and TNn indices are
opposite with the cold bias known with RegCM4 in mean climate simulation (Koné and al.
2018, Klutse and al. 2016). It is very difficult to know the origin of RCM temperature biases,
as they can depend of several factors, such as surface energy fluxes and water, cloudiness,
surface albedo (Sylla et al. 2012; Tadross et al. 2006).
Figure 18 (second panel) is the same as in Fig.14 (second panel), but displays changes in TNn
index. The impact on TNn index of anomalies in initial soil moisture is homogeneous over the
Sahara region, i.e. the wet experiments lead to an increase of TNn index values while the dry
experiments favor a decrease of TNn index values. We noticed this homogeneous impact
coincides with the area of highest TNn index values. However, over the central and west Sahel,



both dry and wet experiments lead to a dominant decrease. Conversely, over the Guinea coast,
we found a dominant increase.
Figure 19 is the same as Fig.15, but shows the PDF distribution of changes in TNn index. The
impact on changes in TNn index, are not homogeneous over all the domains studied. However,
although this impact is weak, over central and west Sahel it tends to decrease, while over the
Guinea coast it tends to increase. For instance, the strongest impact is found over the west Sahel,
where the wet and dry leads to a decrease in TNn index, with peaks at -1°C and -0.2°C
respectively.
In summary, RegCM4 overestimate the TNn index over the entire domain studied. The impact
on TNn index to the soil moisture anomalies is homogeneous only over the Sahara, i.e. the dry
(wet) experiments tend to decrease (increase) the TNn index values. We noticed, this
homogeneous impact coincides with the area of highest TNn index values. However, over the
central and west Sahel, both dry and wet experiments lead to a dominant decrease, while over
the Guinea coast, they lead to a dominant increase.

**3.2.4. The Maximum value of daily minimum temperature (TNx)**
In this section, we turn our attention on the maximum values of daily maximum temperature
(TNx index) for JJAS 2003 and JJAS 2004.  Figure 20 (first panel) is the same as in Fig.14
(first panel), but for TNx index. GTS observation (Fig.20 a, d) shows the maxima of TNx index
values over the Sahara reaching up 40° C, while the minima values reaching 24°C are located
over the Guinea coast sub-region. The reanalysis of EIN (Fig.20 b, e) shows a similar large
scale patterns with PCC value reaching 0.99, but some biases can be noticed between GTS and
EIN datasets. The reanalysis of the EIN underestimates the maxima (not exceeding 38°C) and
the minima (less than 22°C) located respectively over the Sahara and the orographic regions
such as Cameroon mountains, Jos plateau and Guinea highlands. The strongest negative mean
bias is located over the Guinea coast with MB about -3.11 and -3.14°C (resp. JJAS 2003 and
JJAS 2004, Table 3).
As with previous temperature indices, the control experiments (Fig.20 c, f) well reproduce the
general features of TNx index with a PCC value reaching 0.99, but do exhibited some
differences at the local scale. In contrast to the TNN index, the control experiments
underestimate the TNx index, over most of the domains studied. The maxima of TNx index
values are quite underestimate over the Sahara.  For instance, over the central Sahel, the MB is





about -3.85 and -3.99°C (resp. for JJAS 2003 and JJAS 2004, Table 3). This underestimation
of TNx seems to be systematic related to the cold bias in RegCM4 over West Africa which is
shown by several papers (Koné and al. 2018, Klutse and al. 2016).
Figure 20 (second panel) is the same as Fig.14, but displays changes in TNx index, as in Fig.14
(second panel). As for TNn index, the impact on TNx index of anomalies in initial soil moisture
is somewhat homogeneous over the Sahara, i.e. the dry experiments lead to an increase of TNx
index values while the wet experiments favor a decrease of TNx index values. However over
the central and west Sahel, both wet and dry experiments lead to a dominant decrease, although
in the dry experiment, the signal is rather weak.  Conversely, over the Guinea coast, the impact
on TNx index tends to cause a dominant increase.
Figure 21 is the same as Fig.15, but displays the PDF distributions of the changes in TNx index.
As with TNn index, the impact on TNx index changes, is not homogeneous over the entire
domains studied. We noticed that TNX index is more sensitive to the wet and dry experiments
over the central Sahel than the other sub-regions studied. The strongest impact in the wet
experiments, is found over the central Sahel (Fig. 21 a) and it's about -1.3°C, while in dry
experiments it's found over the west Sahel more than -1°C (Fig. 21 b).
In summary, RegCM4 underestimates the TNx index values over the entire domain studied. As
for TNn index, the impact on TNx index to the soil moisture anomalies is homogeneous only
over the Sahara, i.e. the dry (wet) experiments tend to decrease (increase) the TNn index values.
However, over the central and west Sahel, both dry and wet experiments lead to a decrease,
while over the Guinea coast, this impact tends to cause a dominant increase. As compared to
TNn index, the impact on TNx index of the anomalies in initial soil moisture is stronger.
Overall, anomalies in initial soil moisture unequally affect the daily maximum and minimum
temperature over the West African domain. A strong impact is found on daily maximum
temperature extremes than the daily minimum temperature extremes. These results are in line
with the previous works (Jaeger and Seneviratne, 2011; Zhang et al., 2009).
**4. Summary and conclusions**
The impact on the subsequent summer extreme climate of the anomalies in initial soil moisture
over West Africa is investigated using the RegCM4-CLM45. In addition, the performance of
RegCM4-CLM4.5 in representing six extreme indices of precipitation and four extreme indices
of temperature over West Africa was also evaluated. Results have been presented for the two
summers, JJAS 2003 (wet year) and JJAS 2004 (dry year). We performed a sensitivity studies





over the West African domain, with 25 km of spatial resolution. We initialized the control runs
by ERA20C reanalysis soil moisture, and at its wilting points and the field capacity respectively
for dry and wet experiments.
Compared to the extreme indices of the observation datasets, the model overestimated and
underestimated the number of wet days occurrence with respectively a low (1mm.day$^{-1}$) and
high threshold rain rate (e.g 10 mm/day, not shown here). RegCM4 also underestimated the
simple precipitation intensity index (SDII), the maximum 1-day precipitation (Rx1day) and the
total precipitation due to very heavy precipitation days (R95pTOT). The current physical
parameterization scheme of the RegCM4 model results in a positive bias for the number of wet
days with a low precipitation threshold (e. g. 1 mm.day$^{-1}$), while for the indices of number of
wet days with a higher precipitation threshold (e. g. 10 mm.day$^{-1}$, not shown here), it results in
a negative bias. However, the CWD and CDD indices were generally overestimated over the
whole West African domain. On the other hand, the model RegCM4 overestimated the
temperature extreme indices used in this study (TXx, TXn and TNn), except for TNx index,
which is underestimated over the West African domain. As a result, temperature events are
more extreme in the control experiments, except in TNx index.
The impact on extreme precipitation indices of anomalies in initial soil moisture, especially
over the central Sahel, are homogeneous, i.e. dry (wet) experiments tend to decrease (increase)
precipitation extreme indices only for precipitation indices related to the number of
precipitation events (R1mm, CDD and CWD indices), not for those related to the intensity of
precipitation events (SDII, RX1day and R95pTOT indices). Therefore, these results confirm
the strong coupling of land and atmosphere in areas between wet and dry climate regimes (e.g.
Zhang et al., 2011; Koster et al., 2006). In the west Sahel sub-region, the impact of soil moisture
anomalies is homogeneous only for the CDD index, i.e. dry (wet) experiments lead to an
increase (decrease) in the CDD index. While dry and wet experiments result in an increase in
the R1mm, CWD and R95pTOT indices. In the Guinea coast, dry and wet experiments tend to
cause an increase in CWD, R1mm and R95pTOT, except for the CDD index, where they cause
a decrease. We noted that the impact on extreme precipitation indices of anomalies in initial
soil moisture is homogeneous only for indices related to the number of precipitation events
(R1mm, CDD and CWD indices), and not for those related to the amount of precipitation per
event (SDII, RX1day and R95pTOT). It is also important to note that dry and wet experiments





amplify very heavy precipitation days (R95pTOT index) over most of the domain studied. In
addition, among all the precipitation indices studied, the year's contrast has a significant impact
only for the CDD index on the central Sahel for wet experiments.
The impact on extreme temperatures of anomalies in initial soil moisture is generally greater
than on extreme precipitation. Initial soil moisture anomalies unequally affect daily minimum
and maximum temperature. A strong impact is found on maximum temperature than minimum
temperature. Wet (dry) experiments result in an increase (decrease) in the TXx and TXn indices
in most of the areas studied. Contrary to the indices related to the maximum temperature (TXx
and TXn), the impact of soil moisture on the indices related to the minimum temperature (the
TNx and TNn indices) is not homogeneous over most of the domains studied. The strongest
impacts on minimum temperature indices are found over the Sahara where the TNn and TNx
indices values are higher and their changes are somewhat homogeneous. In fact, initial moisture
anomalies in dry (wet) soils tend to cause an increase (a decrease) in the TNn and TNx indices
over the Sahara. However, in west and central Sahel, both dry and wet experiments tend to
decrease the TNn and TNx indices, but increase them over the Guinea coast.
Overall, the impact on precipitation of the anomalies in initial soil moisture is much more
complicated, as compared to temperature. For a proper assessment of the dependence of the
model in our results, it would be appropriate to repeat the investigation using different RCMs
in a multi-model framework.
**Author contribution**
The authors declare to have no conflict of interest with this work. B. Koné and A. Diedhiou
fixed the analysis framework. B. Koné carried out all the simulations and figures production
according to the outline proposed by A. Diedhiou. B. Koné and A. Diedhiou, S. Anquetin and
A. Diawara worked on the analyses. All authors contributed to the drafting of this manuscript.
**Acknowledgements**
The research leading to this publication is co-funded by the NERC/DFID "Future Climate for
Africa" programme under the AMMA-2050 project, grant number NE/M019969/1 and by
IRD (Institut de Recherche pour le Développement; France) grant number UMR IGE
Imputation 252RA5.



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





**TABLES AND FIGURES.**

| | Extreme indices | Definition | Units |
|---|---|---|---|
| | Extreme Rainfall Indices | | |
| 1 | R1mm | count of days when daily precipitation ≥1mm | day |
| 2 | SDII | total precipitation divided by total number of rain days with daily precipitation above 1mm | mm/day |
| 3 | CDD | maximum length of dry spell, maximum number of consecutive days with R < 1 mm day −1 | day |
| 4 | CWD | maximum length of wet spell, maximum  number of consecutive days with R ≥ 1 mm day −1 | day |
| 5 | RX1day | Maximum 1 day precipitation amount | mm |
| 6 | R95pTOT | Total precipitation due to days with precipitation exceeding the 95th percentiles for wet-day amounts. | mm |
| | Extreme temperature indices | | |
| 7 | TXn | Minimum value of daily maximum temperature | °C |
| 8 | TXx | Maximum value of daily maximum temperature | °C |
| 9 | TNn | Minimum value of daily minimum temperature | °C |
| 10 | TNx | Maximum value of daily minimum temperature | °C |


**Table1**: The 10 extreme climate indices used in this study.








|  |  | Central Sahel | | West Sahel | | guinea | | West Africa | |
|---|---|---|---|---|---|---|---|---|---|
|  |  | MB | PCC | MB | PCC | MB | PCC | MB | PCC |
| R1mm | TRMM_2003 | -6.76 | 0.98 | -3.15 | 0.99 | 8.89 | 0.99 | -1.12 | 0.98 |
| | CTRL_2003 | 33.17 | 0.98 | -5.25 | 0.96 | 53.16 | 0.96 | 22.18 | 0.96 |
| | TRMM_2004 | -7.51 | 0.98 | -3.42 | 0.99 | 10.44 | 0.98 | -1.34 | 0.98 |
| | CTRL_2004 | 29.50 | 0.98 | 1.34 | 0.96 | 55.46 | 0.96 | 23.85 | 0.95 |
| SDII | TRMM_2003 | 2.67 | 0.96 | 0.22 | 0.94 | -5.24 | 0.95 | 1.20 | 0.86 |
| | CTRL_2003 | -7.52 | 0.97 | -9.95 | 0.94 | -13.62 | 0.77 | -7.67 | 0.73 |
| | TRMM_2004 | 2.07 | 0.96 | 0.45 | 0.96 | -6.44 | 0.94 | 1.16 | 0.86 |
| | CTRL_2004 | -7.01 | 0.97 | -9.37 | 0.94 | -14.65 | 0.81 | -7.59 | 0.77 |
| CDD | TRMM_2003 | 1.21 | 0.95 | 0.89 | 0.93 | -0.93 | 0.94 | -2.29 | 0.92 |
| | CTRL_2003 | 0.93 | 0.90 | 14.49 | 0.91 | -7.84 | 0.66 | 2.63 | 0.85 |
| | TRMM_2004 | 2 | 0.95 | 1.58 | 096 | -3.17 | 0.92 | -1.75 | 0.94 |
| | CTRL_2004 | 4.75 | 0.91 | 17.51 | 0.95 | -9.43 | 0.68 | 6.99 | 0.89 |
| CWD | TRMM_2003 | -0.48 | 0.92 | 0.80 | 0.94 | 2.47 | 0.92 | 0.37 | 0.90 |
| | CTRL_2003 | 45.56 | 0.83 | 18.44 | 0.75 | 59.21 | 0.88 | 31.20 | 0.81 |
| | TRMM_2004 | -0.68 | 0.92 | 0.97 | 0.92 | 2.38 | 0.89 | 0.26 | 0.87 |
| | CTRL_2004 | 36.78 | 0.79 | 20.48 | 0.78 | 60.51 | 0.82 | 29.74 | 0.79 |
| RX1day | TRMM_2003 | 35.78 | 0.92 | 25.31 | 0.89 | 14.31 | 0.86 | 26.02 | 0.84 |
| | CTRL_2003 | -26.46 | 0.78 | -38.07 | 0.91 | -30.28 | 0.54 | -20.08 | 0.50 |
| | TRMM_2004 | 31.66 | 0.91 | 20.19 | 0.91 | 10 | 0.88 | 22.19 | 0.85 |
| | CTRL_2004 | -22.89 | 0.46 | -36.67 | 0.88 | -42.44 | 0.42 | -20.23 | 0.40 |
| R95pTOT | TRMM_2003 | 23.19 | 0.92 | 13.31 | 0.94 | -0.23 | 0.96 | 16.54 | 0.91 |
| | CTRL_2003 | -27.67 | 0.67 | -33.39 | 0.77 | -43.22 | 0.65 | -29.12 | 0.59 |
| | TRMM_2004 | 23.26 | 0.91 | 12.32 | 0.94 | -0.93 | 0.95 | 18.54 | 0.91 |
| | CTRL_2004 | -24.38 | 0.46 | -31.75 | 0.80 | -46.61 | 0.60 | -27.45 | 0.55 |


**Table 2**: The pattern correlation coefficient (PCC) and the mean bias (MB) of R1mm (in day),
SDII (in mm/day), CDD (in day), CWD (in day), RX1day (in mm) and R95pTOT (in mm)
indices for TRMM observation and their corresponding control experiments (initialized with
initial soil moisture of ERA20C reanalysis) with respect to CHIRPS, calculated over Guinea
coast, central Sahel, west Sahel and the entire West African domain for JJAS 2003 and JJAS

878 2004.







|  |  | Central Sahel | | West Sahel | | guinea | | West Africa | |
|---|---|---|---|---|---|---|---|---|---|
|  |  | MB | PCC | MB | PCC | MB | PCC | MB | PCC |
| TXx | TRMM_2003 | -2.17 | 0.99 | -3.05 | 0.99 | -4 | 0.99 | -2.77 | 0.99 |
|  | CTRL_2003 | 2.10 | 0.99 | 3.02 | 0.99 | -1.34 | 0.99 | 0.32 | 0.99 |
|  | TRMM_2004 | -2.44 | 0.99 | -3.86 | 0.99 | -3.84 | 0.99 | -2.94 | 0.99 |
|  | CTRL_2004 | 1.14 | 0.99 | 2.02 | 0.99 | -1.41 | 0.99 | -0.16 | 0.99 |
| TXn | TRMM_2003 | 0.31 | 0.99 | -1.48 | 0.99 | -0.70 | 0.99 | 0.50 | 0.99 |
|  | CTRL_2003 | 5.12 | 0.99 | 6.56 | 0.99 | 3.76 | 0.99 | 5.65 | 0.99 |
|  | TRMM_2004 | -0.76 | 0.99 | -1.73 | 0.99 | -1.38 | 0.99 | -0.32 | 0.99 |
|  | CTRL_2004 | 3.43 | 0.99 | 5.44 | 0.99 | 2.75 | 0.99 | 4.14 | 0.99 |
| TNn | TRMM_2003 | 3.08 | 0.99 | 3.43 | 0.99 | 1.28 | 0.99 | 3.15 | 0.99 |
|  | CTRL_2003 | 2.37 | 0.99 | 3.30 | 0.99 | 1.53 | 0.99 | 1.45 | 0.99 |
|  | TRMM_2004 | 3.28 | 0.99 | 2.98 | 0.99 | 1.20 | 0.99 | 3.11 | 0.99 |
|  | CTRL_2004 | 2.09 | 0.99 | 2.55 | 0.99 | 1.28 | 0.99 | 0.71 | 0.99 |
| TNx | TRMM_2003 | -0.69 | 0.99 | -1.79 | 0.99 | -3.11 | 0.99 | -1.62 | 0.99 |
|  | CTRL_2003 | -1.91 | 0.99 | -2.86 | 0.99 | -3.35 | 0.99 | -3.85 | 0.99 |
|  | TRMM_2004 | -0.82 | 0.99 | -1.43 | 0.99 | -3.14 | 0.99 | -1.71 | 0.99 |
|  | CTRL_2004 | -1.90 | 0.99 | -2.54 | 0.99 | -3.32 | 0.99 | -3.99 | 0.99 |


**Table 3**: The pattern correlation coefficient (PCC) and the mean bias (MB in°C) of TXx,
TXn, TNn and TNx indices from the reanalyze of EIN and their corresponding control
experiments (initialized with initial soil moisture of ERA20C reanalysis) with respect to GTS,
calculated for Guinea coast, central Sahel, west Sahel and the entire West African domain for
JJAS 2003 and JJAS 2004.





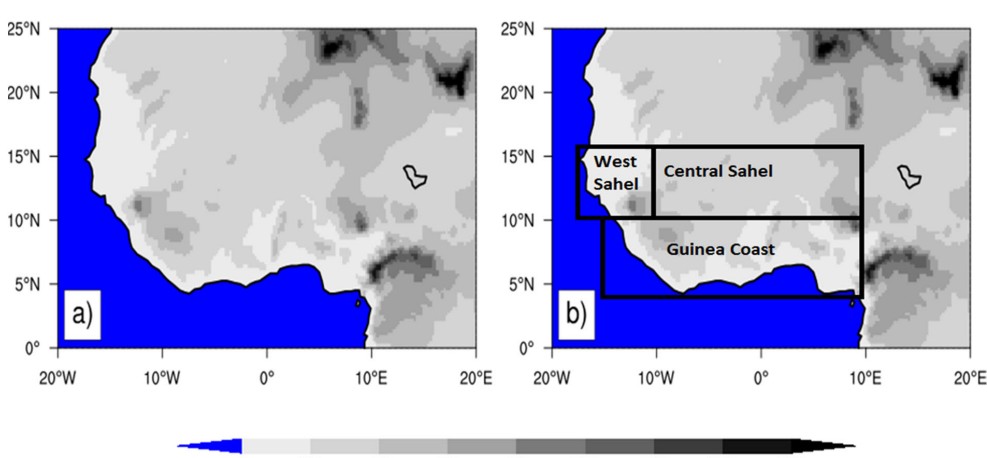



**Figure 1:** Topography of the West African domain. The analysis of the model result has an emphasis on the whole West African domain and the three subregions Guinea coast, central Sahel and west Sahel, which are marked with black boxes.




















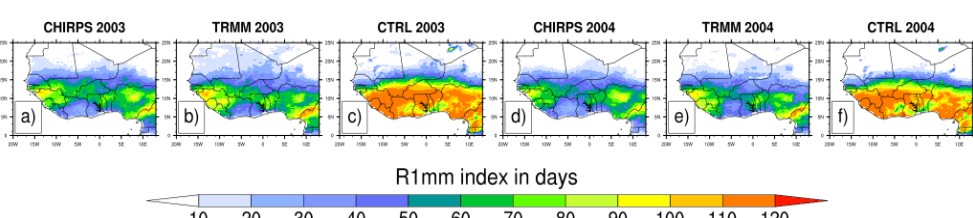

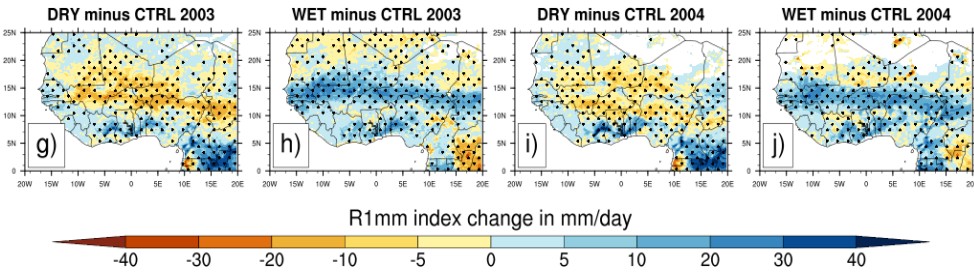



**Figure2:** Observed 4-month averaged (JJAS) mean values of wet days occurrence (R1mm index in days) from CHIRPS (a and d) and TRMM(b and e) observations for JJAS 2003 and JJAS 2004 and their corresponding simulated control (CTRL) experiments (c and f) initialized with initial soil moisture of the reanalysis of ERA20C (first panel) and changes in R1mm index in days (second panel) for JJAS 2003 and JJAS 2004, from dry (g and i) and wet (h and j) experiments with respect to the corresponding control experiments. Areas with values passing the 10% significance test are dotted.

















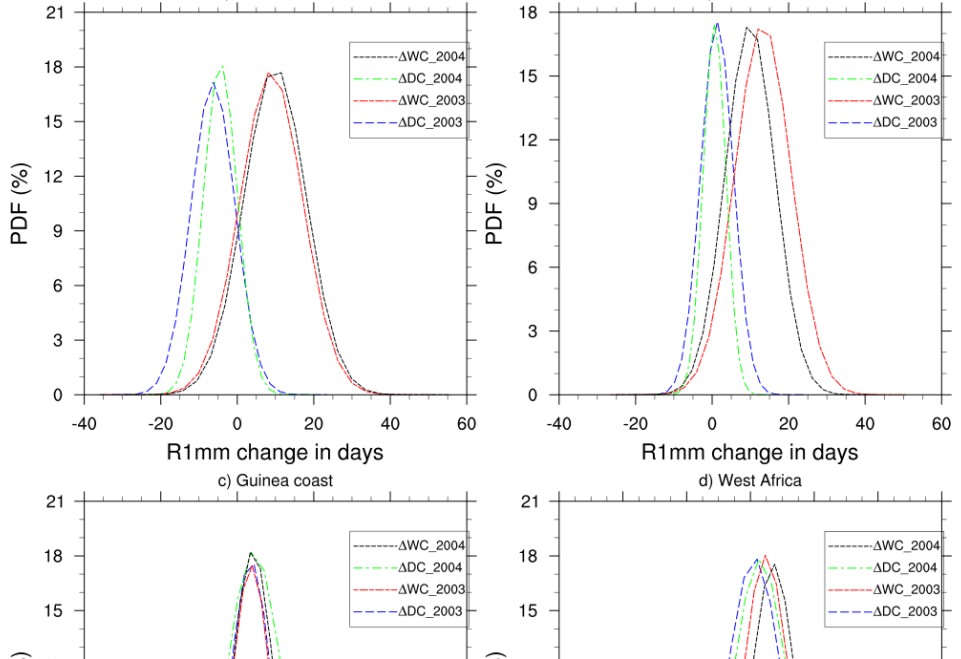



**Figure3:** PDF distributions (%) of mean values of wet days occurrence change in JJAS 2003
and JJAS 2004, over (a) central Sahel , (b) West Sahel, (c) Guinea and (d) West Africa derived
from dry (ΔDC) and wet (ΔWC) experiments with respect to their corresponding control
experiment.









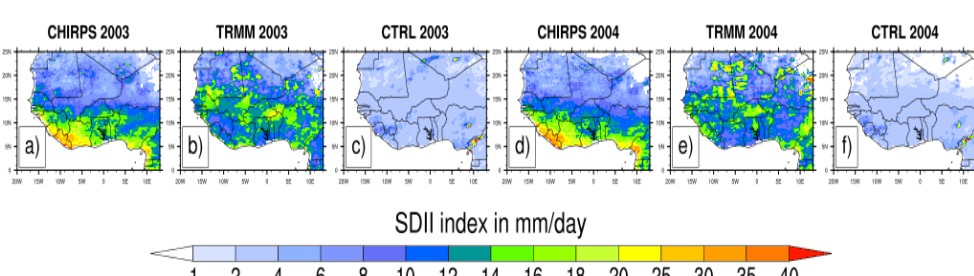

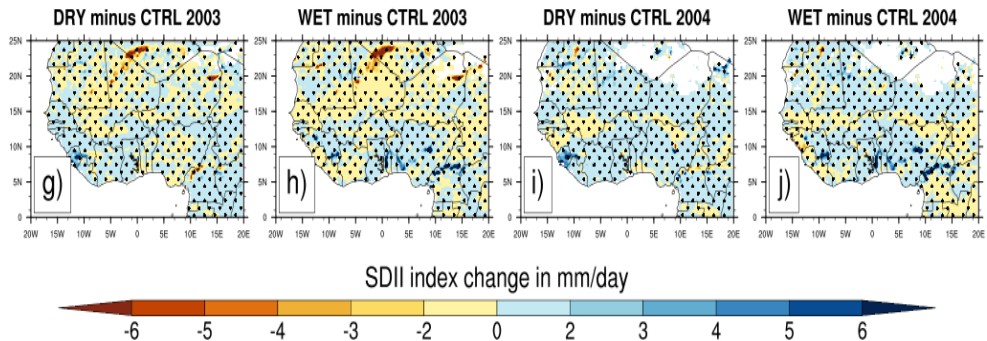



**Figure4:** Same as Fig. 2 but for the SDII index (in mm/day).




















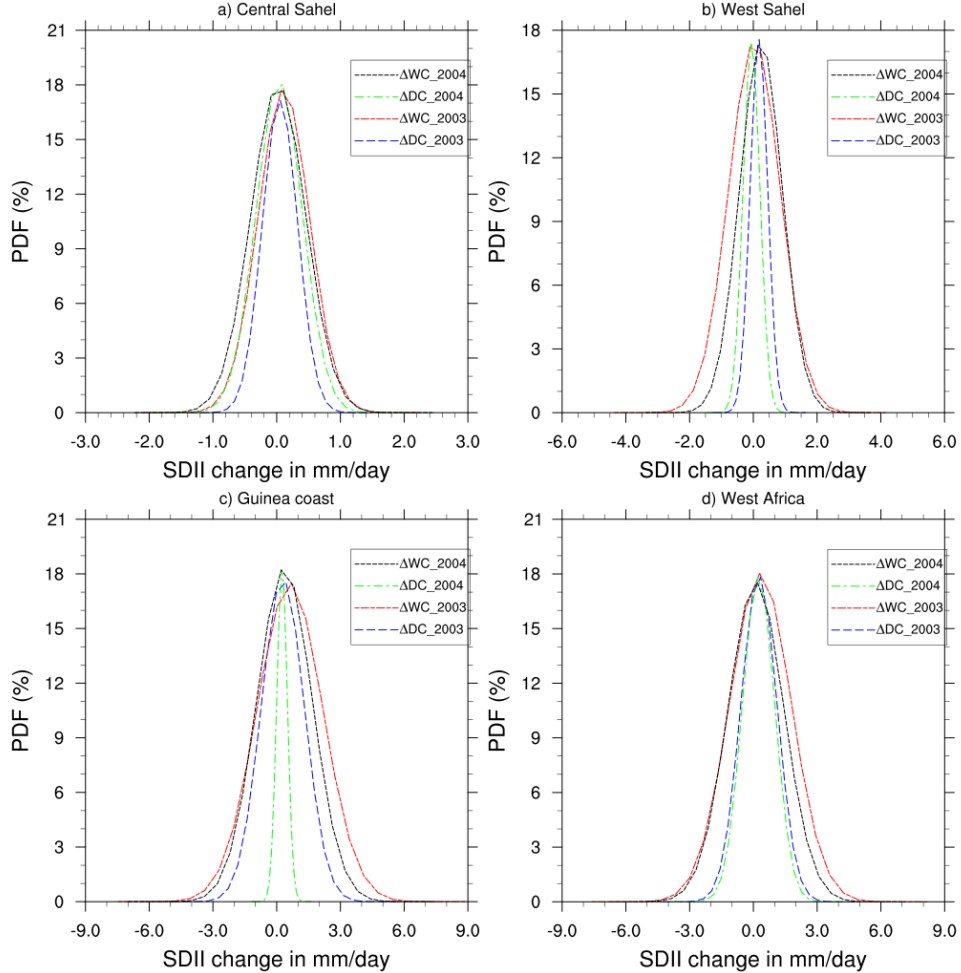




**Figure 5:** Same as Fig. 3 but for the SDII index (in mm/day).












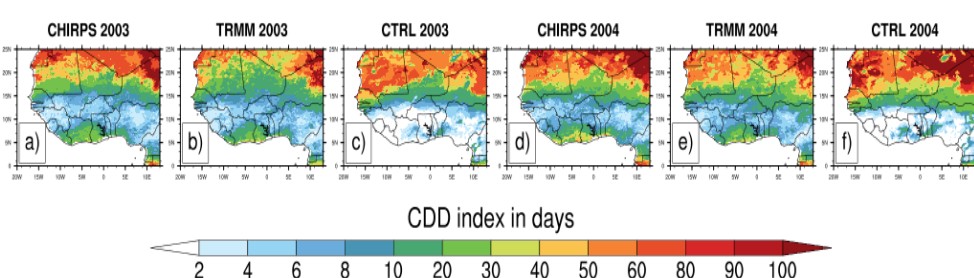

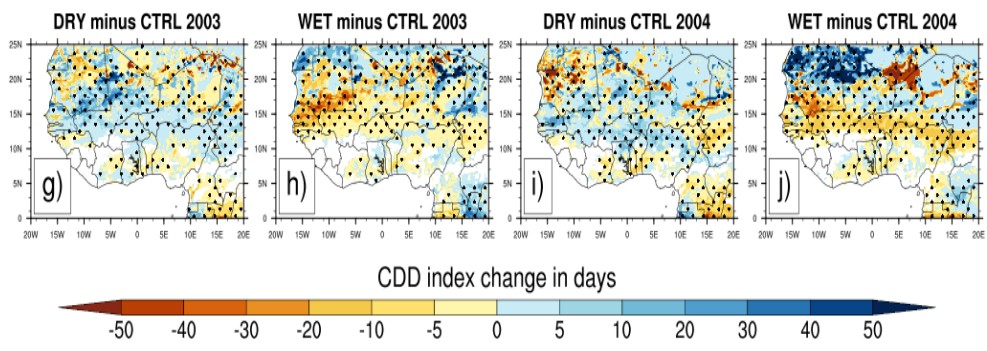



**Figure 6**: Same as Fig. 2 but for the CDD index (in day).



















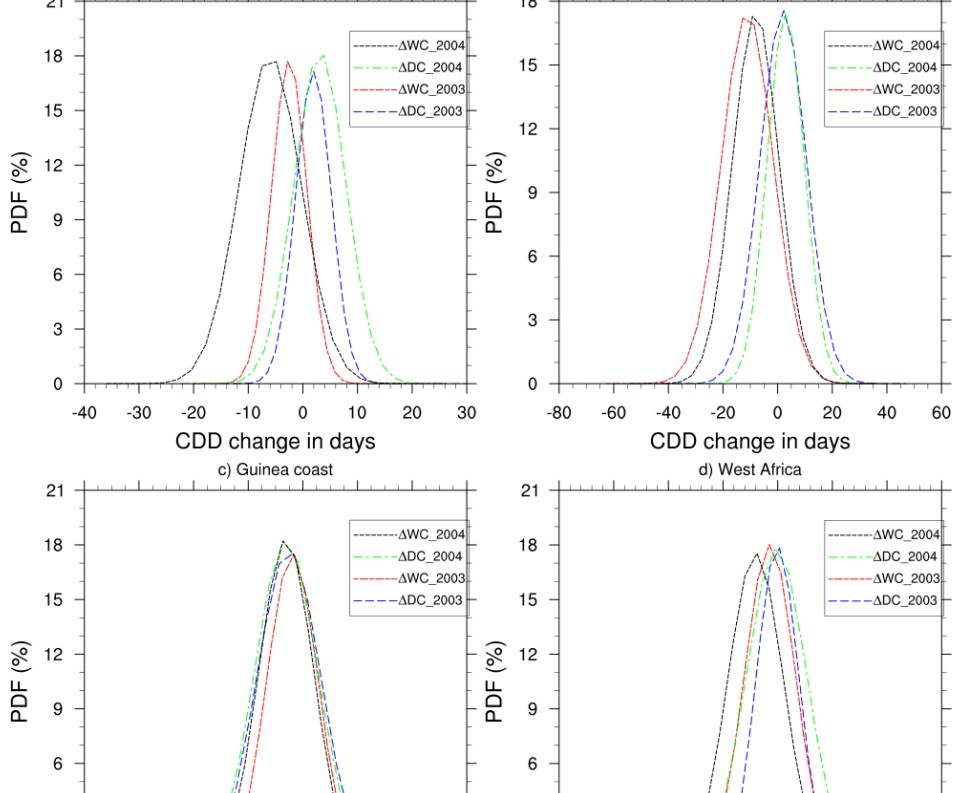




**Figure 7:** Same as Fig. 3 but for the CDD index (in day).








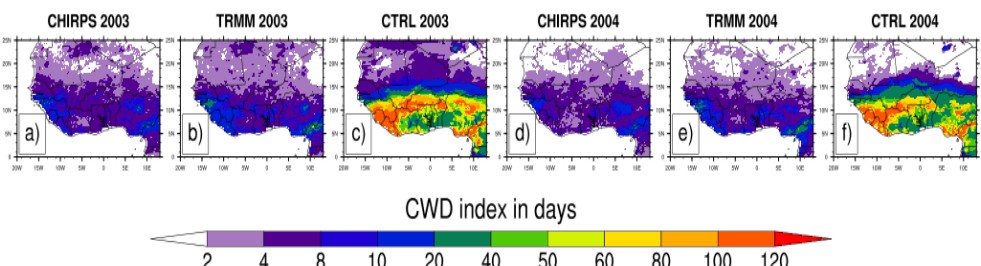

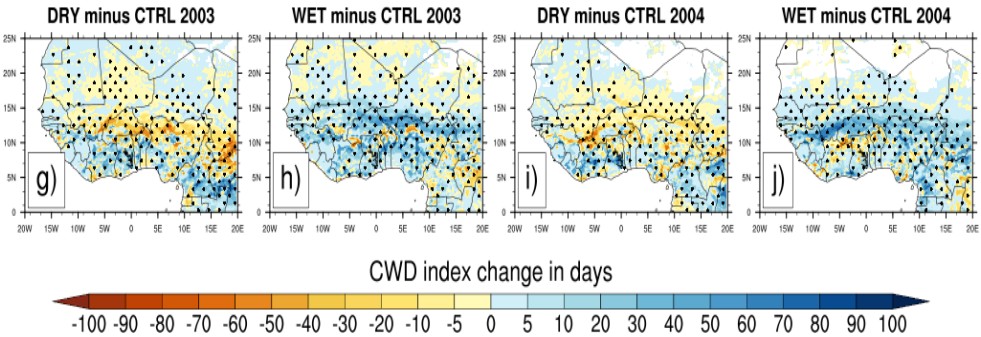

**Figure 8:** Same as Fig. 2 but for the CWD index (in day).








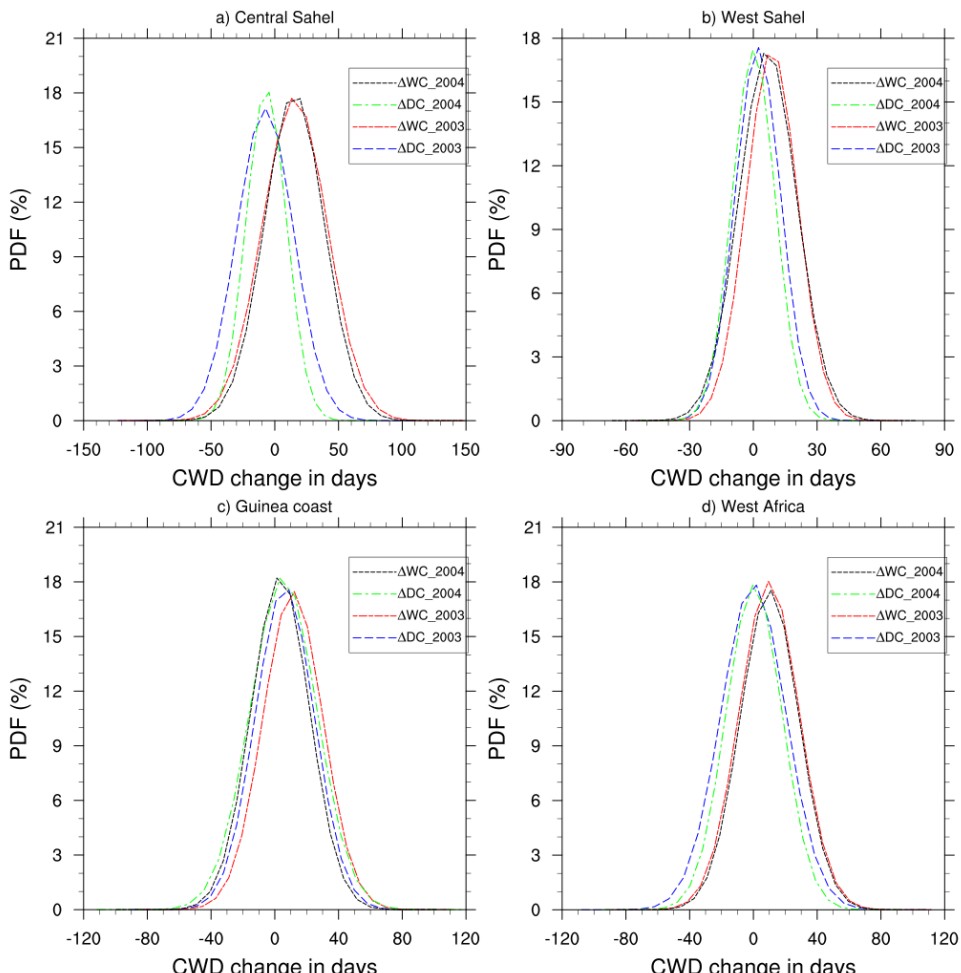




**Figure 9**: Same as Fig. 3 but for the CWD index (in day).












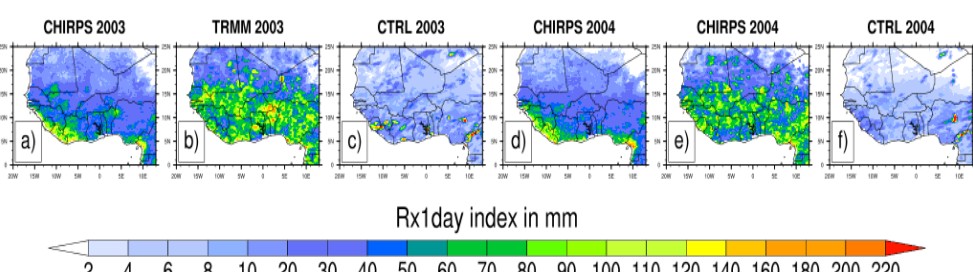

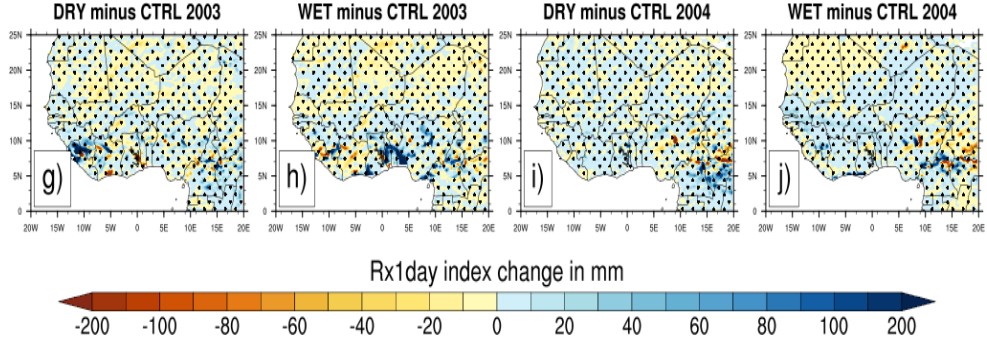



**Figure 10**: Same as Fig. 2 but for the RX1day index (in mm).


















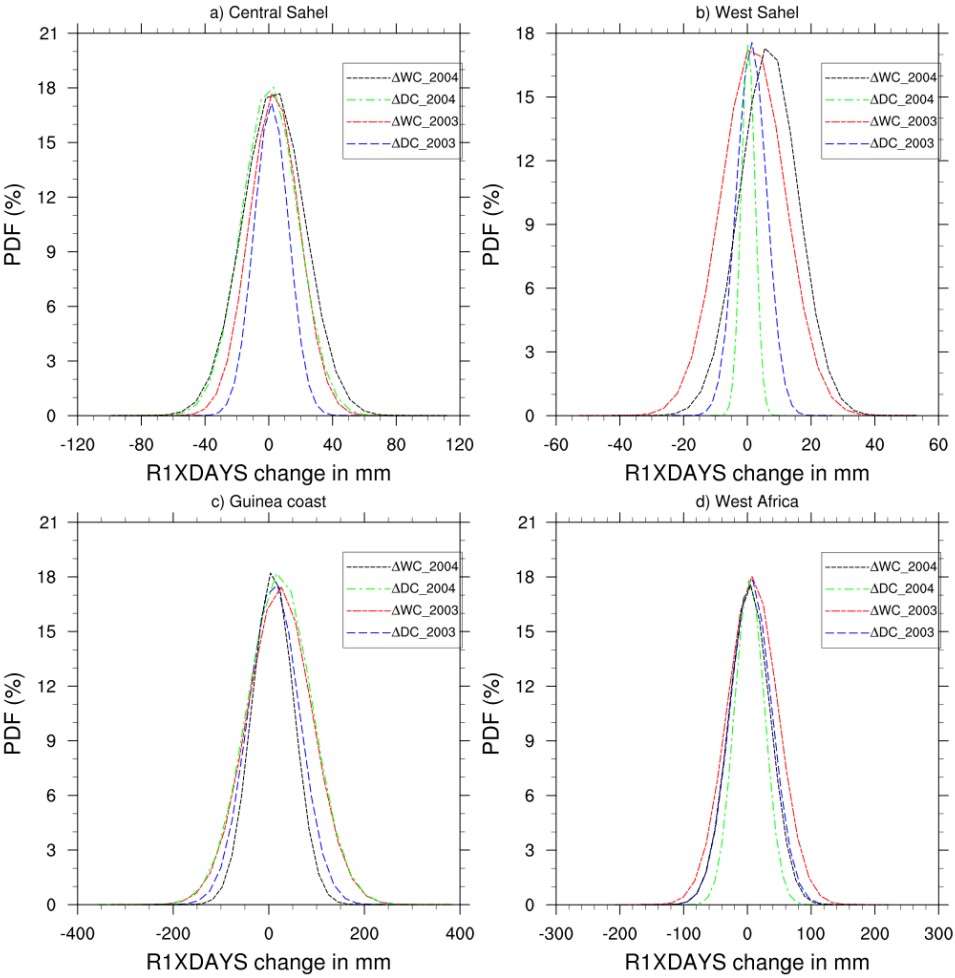



**Figure 11**: Same as Fig. 3 but for the RX1DAY index (in mm).











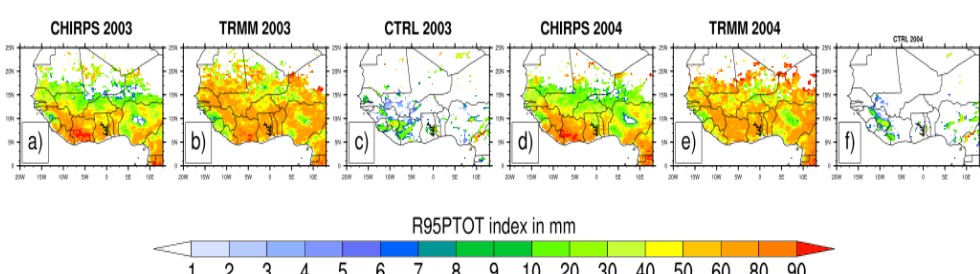

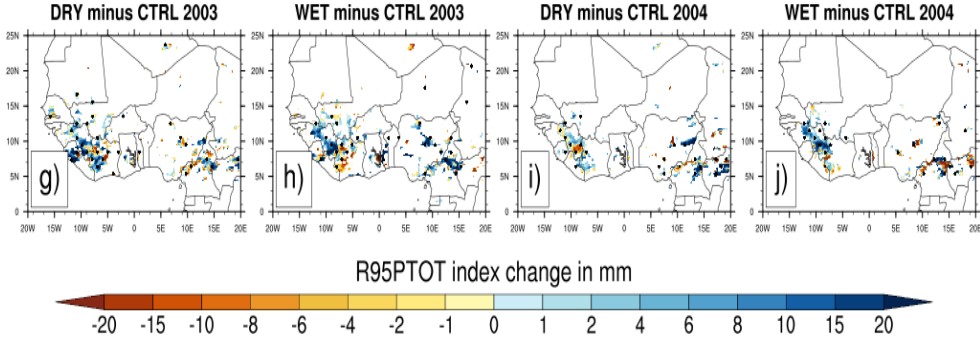


**Figure 12:** Same as Fig. 2 but for the R95pTOT index (in mm).























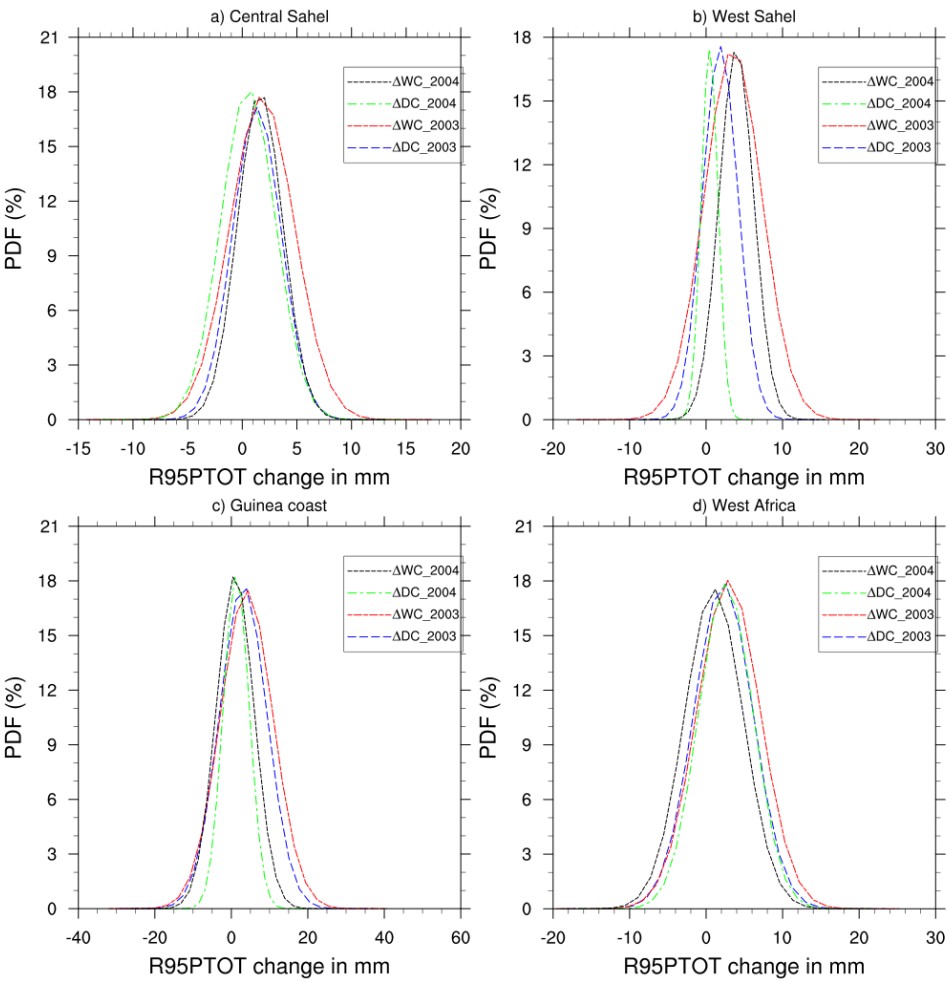


**Figure 13:** Same as Fig. 3 but for theR95pTOT index (in mm).









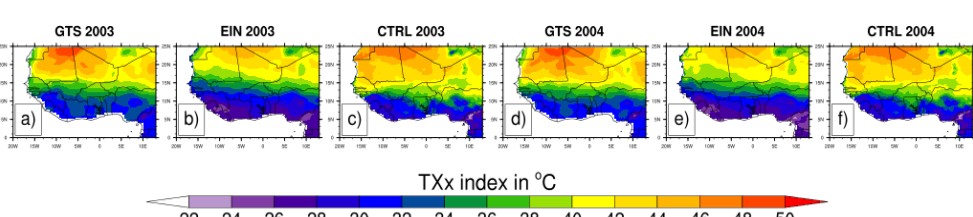

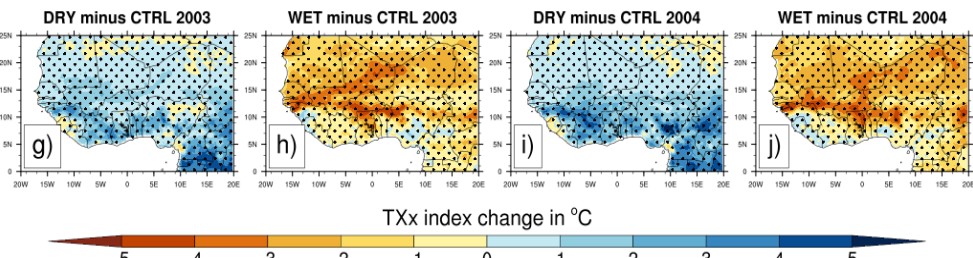

**Figure 14:** Observed 4-month averaged (JJAS) maximum value of daily maximum temperature (TXx index in°C) from GTS observation (a and d) and the reanalysis of EIN (b and e) for JJAS 2003 and JJAS 2004 and their corresponding simulated control (CTRL) experiments (c and f) initialized with the initial soil moisture of the ERA20C reanalysis (first panel) and changes in TXx index in°C (second panel) for JJAS 2003 and JJAS 2004, from dry (g and i) and wet (h and j) experiments with respect to the corresponding control experiments. Areas with values passing the 10% significance test are dotted.



























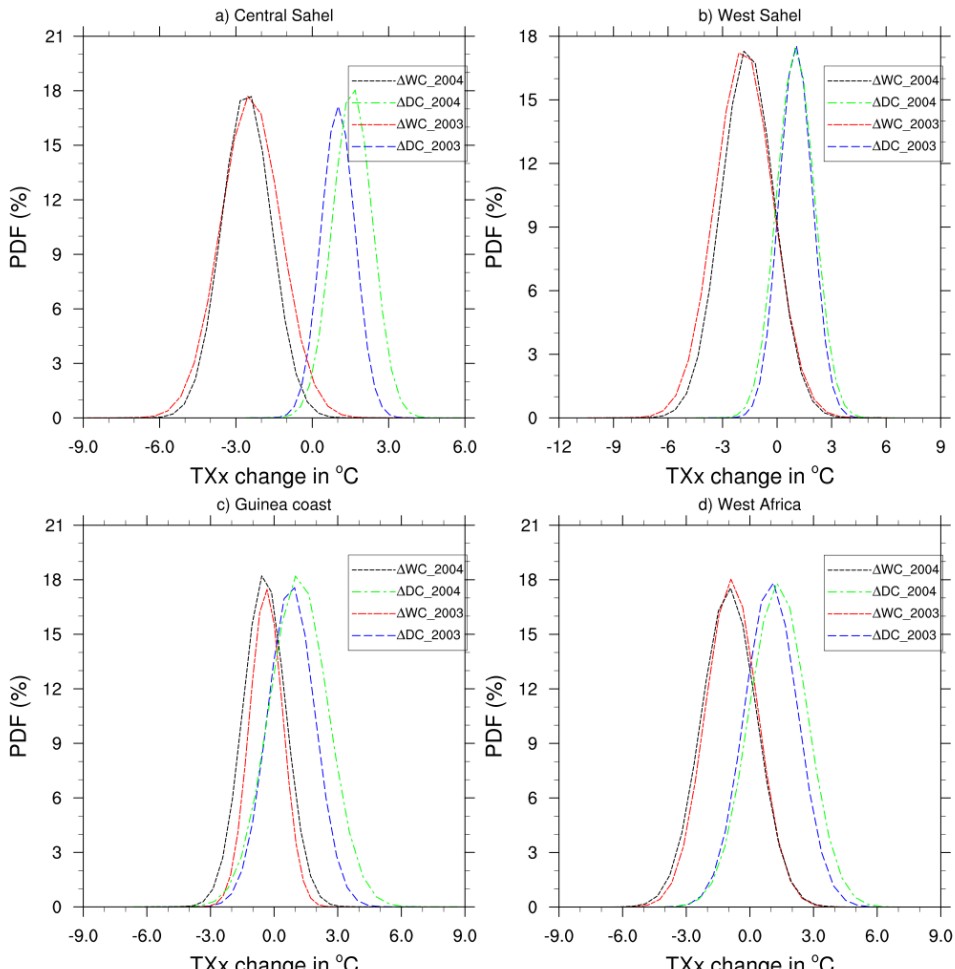



**Figure 15:** PDF distributions (%) of change in maximum value of daily maximum temperature
(TXx index, in°C)  for JJAS 2003 and JJAS 2004,  over (a) central Sahel , (b) West Sahel, (c)
Guinea and (d) West Africa derived from dry (ΔDC) and wet (ΔWC) experiments compared to
their corresponding control experiment.









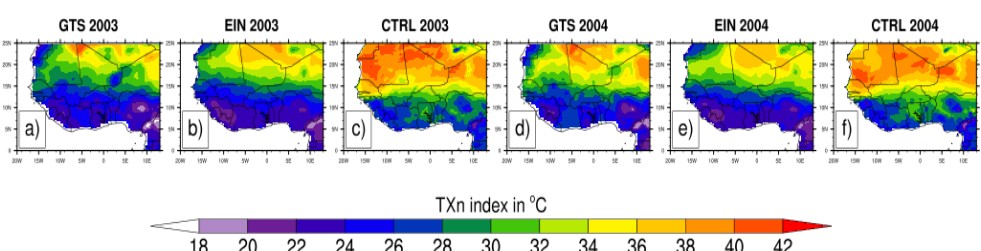

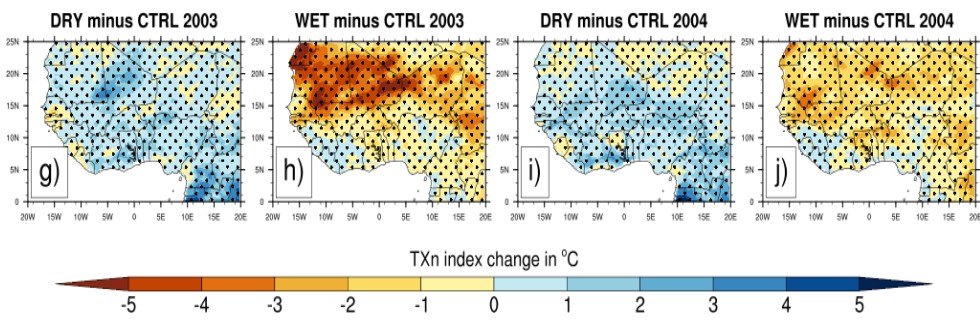




**Figure 16:** Same as Fig. 14 but for the TXn index




















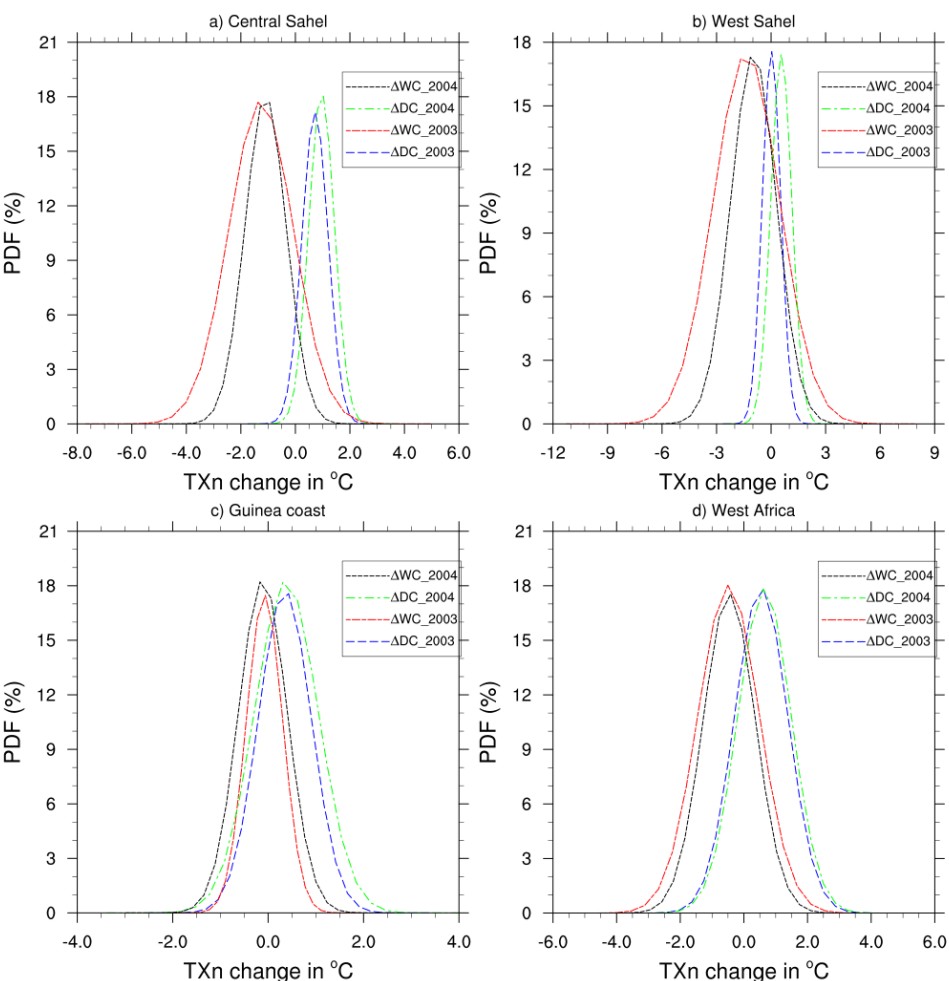



**Figure 17:** Same as Fig. 15 but for the TXn index.












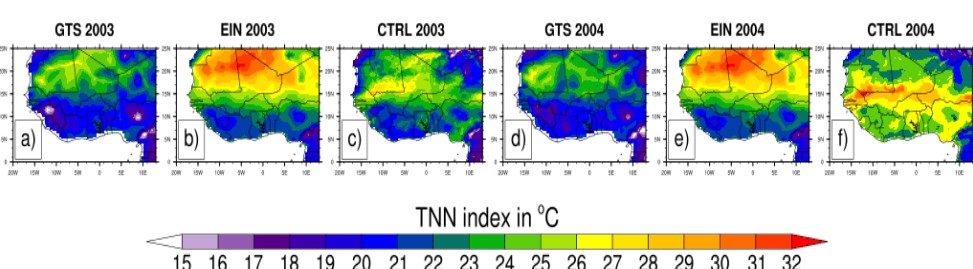

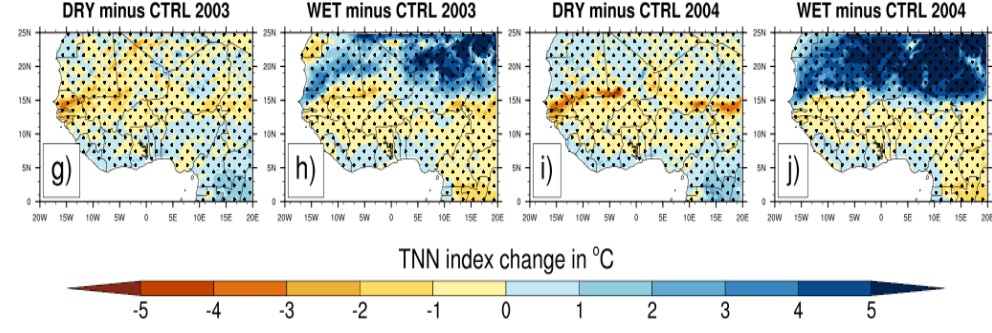




**Figure 18:** Same as Fig. 14 but for the TNn index.



















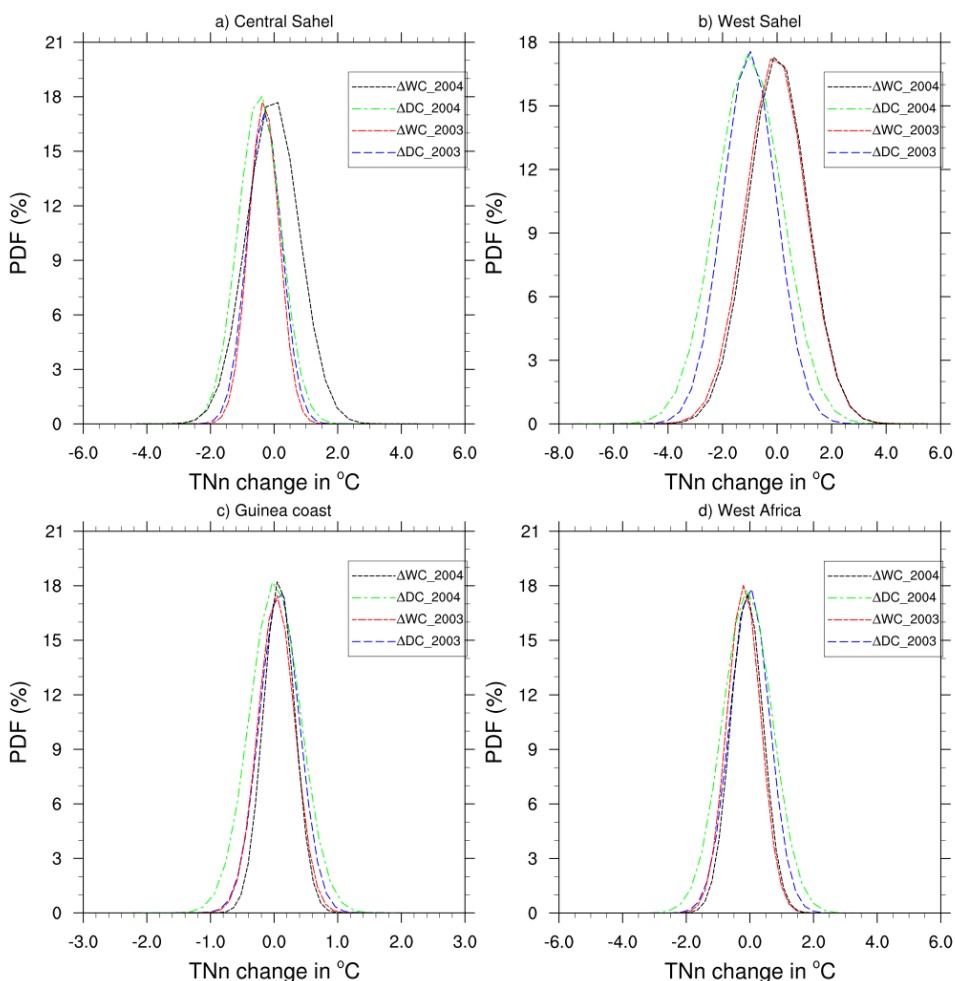




**Figure 19:** Same as Fig. 14 but for the TNn index.






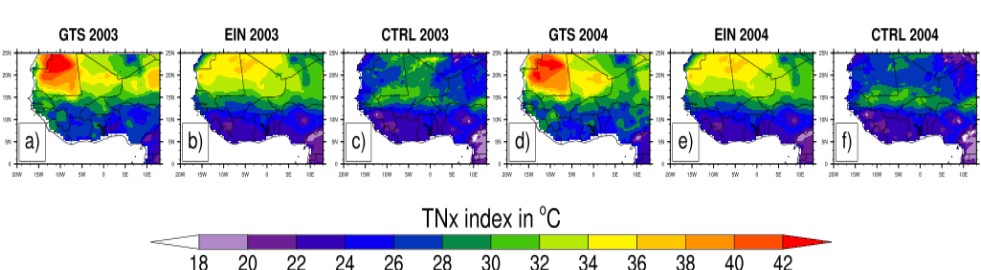

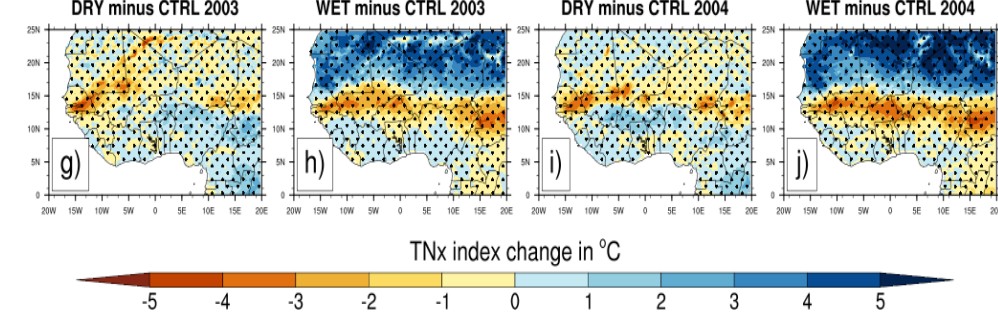

**Figure 20:** Same as Fig. 14 but for the TNx index








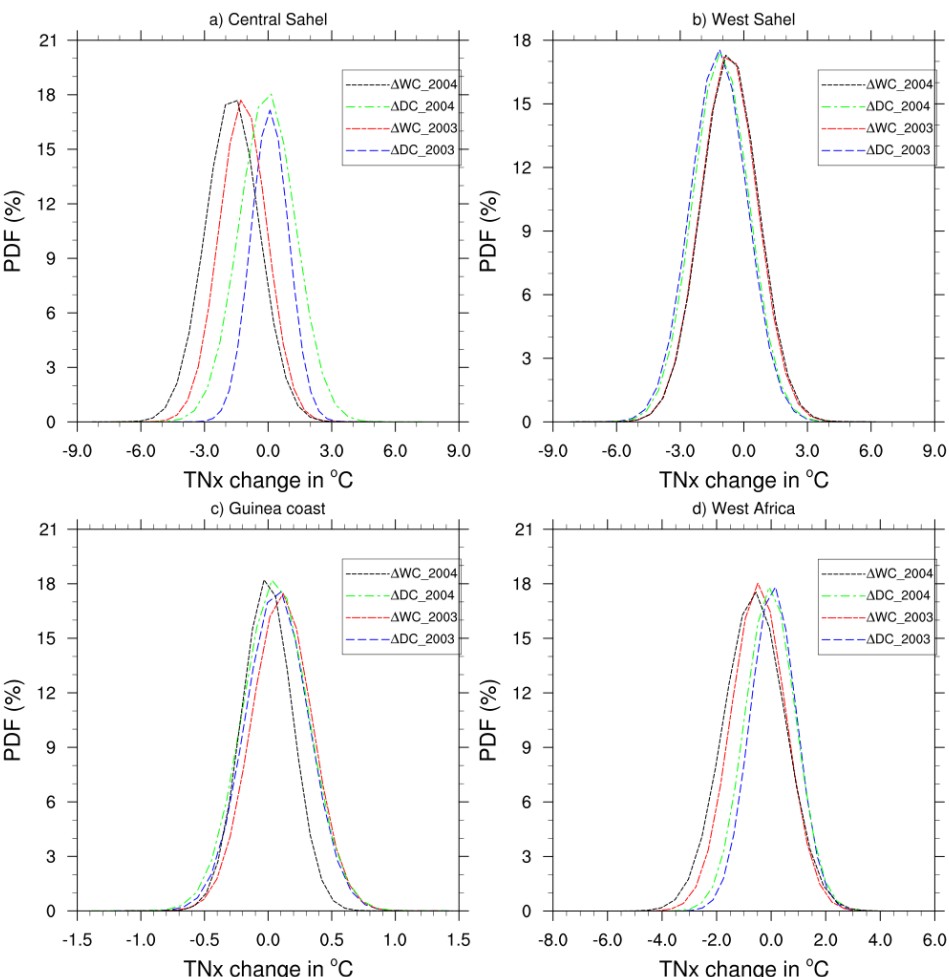



**Figure 21:** Same as Fig. 15 but for the TNx index.
