# Peer review of "Influence of initial soil moisture in a Regional Climate Model study over West Africa. Part 2: Impact on the climate extremes."

_Hydrology and Earth System Sciences, 2020_

## Referee Comment (RC1) · Anonymous Referee #1 · 17 Jun 2020

Reviewer Comments: Influence of initial soil moisture in a Regional Climate Model study over West Africa. Part 2: Impact on the climate extremes.

This manuscript is a follow-up to "Influence of initial soil moisture in a Regional Climate Model study over West Africa: Part 1: Impact on the climate mean", and focuses on changes to extreme indices over West Africa in a variety of experiments. The authors find that dry experiments decrease the number of precipitation events (vice versa for wet events), but does not impact the intensity. Temperature is more significantly impacted, particularly maximum temperature.

I think that the results for extremes will be of great use, however, I will note that I

also reviewed part one of this manuscript, any many of my initial points and concerns about the experiment design still stand. These have been copied below under "major comments". I have some other points and concerns for this manuscript below as well. Overall, I think both manuscripts are in need of major revision.

Major Comments: 1. As noted above, my main concern stems from your year choices. While this results in 6 experiments for comparison, I am not convinced that the results are robust given only a 2 year sample size. Moreover, I'm curious how these years were chosen - are they extreme wet and dry years? How often do years such as these occur? How is "wet" and "dry" defined?

2. Given you only have a few experiments, a significance test would be difficult to achieve. Do you feel that you have sufficiently large "n" to address significance? You may have enough grid points to address spatial significance (given that you modify your n due to autocorrelation, but I note in my minor comments that you should include your sample size), but temporally I don't feel the results are robust.

3. It would be prudent to discuss the implications of this work beyond a summary, perhaps in the concluding remarks.

4. You offer a comparison of CHIRPS and TRMM, and find large differences in the two datasets. How does this impact your results?

5. I have this problem a lot with manuscripts that include extreme indices - there are a huge amount of indices to show, and this adds to length and can cause the reader to get lost in the paper as you go through each one. 21 figures is a lot! I like the way that you have isolated each index, but I think you could cut down on the detail slightly to save some words and not have the reader get lost in the details.

Minor Comments: 1. I noticed a number of grammatical and spelling errors in the manuscript, I suggest having someone read and edit the manuscript specifically for editorial remarks such as these.

2. You use a number of parenthetical references such as "impacts of the wet (dry) soil moisture on wet (dry) years etc." - I do not mind these at all, but sometimes the text is very difficult to read when they are used in excess.

3. Define the lat and lon range of your domain(s).

4. Line 102: Does this contradict your statement on line 23 of part 1? Perhaps rewording is necessary.

5. Line 138 to 147: I believe you're talking about autocorrelation - neighboring grid points are spatially dependent. You do not necessarily need to resample, but you can estimate your n given autocorrelation - sometimes called effective sample size. I think you're using NCL in much of this manuscript (at least, your Figures look like NCL!), which has functions to calculate sample autocorrelation and equivalent sample size.

6. Line 198: "Indicating that the number of wet days occurrence are occurred more likely not only in wet experiments but also in the dry experiments." I do not understand this sentence. 7. I noticed that sometimes your section summaries only include some of your results - is there a way to make these more comprehensive without adding to length?

---

## Referee Comment (RC2) · Anonymous Referee #2 · 24 Jun 2020

This paper by Koné et al. builds on a companion manuscript by evaluating the effect of initial soil moisture conditions on climate extremes. This paper could be of interest to the scientific community, but my previous concerns from the companion paper apply to this manuscript as well. My previous concerns were about the choice of initial soil moisture conditions for the sensitivity experiment and the choice of study years. Below I outline one additional major concern and a few minor concerns.

Model evaluation: The authors need to either demonstrate that the model used can reproduce precipitation or temperature extremes in the study region or provide a citation demonstrating this, otherwise this model may not be a good tool for this research

question. It's important that the evaluation be of precipitation extremes rather than the means or seasonal cycle (as in Koné et al. 2018) since that is what the authors are focusing on.

Minor points: Statistical significance: Perhaps I misunderstood the methods, but it seems like statistical significance can't be evaluated using this model setup (which is okay) but it shouldn't be presented as if it can. Each point only has a control year and two model runs right? Please explain this further, the methods section does not provide enough detail here. What is your null distribution and what is your test distribution at each point?

PDF figures: In my opinion the PDFs don't add information and should probably be removed from both manuscripts to save space. The PDFs duplicate the spatial maps of changes, which provide more information, and double the number of figures presented.

Pattern correlations in Table 3: It's not clear exactly how to interpret the pattern correlations for temperature. A value of 0.99 for every single value seems to imply that either there's an error in the calculation or that the metric is not useful. Are the temperature datasets this closely aligned, and if so would it be more useful to assess pattern correlation of temperature anomalies rather than the absolute temperature? I assume that the labels for TRMM should be EIN here as well.

---

## Author Comment (AC1) · 3 Sep 2020

Major/Specific Comments: 1. Comments: As noted above, my main concern stems from your year choices. While this results in 6 experiments for comparison, I am not convinced that the results are robust given only a 2 year sample size. Moreover, I'm curious how these years were chosen -are they extreme wet and dry years? How often do years such as these occur? How is "wet" and "dry" defined? Author's response: Thank you for your comment. We re-run the simulations over 5 years (2001 to 2005) during the months of June to September over our West African domain. We superimposed the 5 years and their climatological average in order to analyze the changes in

daily soil moisture over our domain studied (Fig.1). The Fig.1 shows that the weakest and strongest impact of the dry experiments is found for 2003 and 2004 respectively. For a wet year, the impact of drying out soil moisture is quickly erased. While for a dry year the impact of the drying of the soil is accentuated. This meaning that 2003 and 2004 are respectively the wettest and driest years in dry experiment. However, for the wet experiments, the weakest impact is found for 2004, and the strongest impact is found for the years 2001, 2002 and 2004. In a dry year, the impact of soil humidification is very quickly erased, while in a wet year the impact of soil humidification is accentuated. The wet experiments confirm the result obtained in dry experiments, 2003 and 2004 are wettest and driest years respectively. To conduct our analyzing to estimate the limits of the impact of internal soil moisture forcing on the new dynamical core non-hydrostatic of RegCM4, we have been used the two extreme years 2003 and 2004 (resp. the wettest and the driest years) among the 5 years. It is in the same context, several previous studies chosen two extreme years for their sensitivity study of initial soil moisture condition on the models. Hong and al. (2000) use in their study only two years (3 months per year) to investigate the impact of initial soil moisture over the North of America (in the Great Plains) during the two summers, May-June-July (MJJ) 1988 (corresponding to a drought in the Great plains) and MJJ 1993 (correspond to a flooding event). Over Asia, Kim and Hong (2006) in their paper "Impact of Soil Moisture Anomalies on Summer Rainfall over East Asia: A Regional Climate Model Study" used two contrasted years 1997 (below normal precipitation year) and 1998 (above normal precipitation year).

2. Comments from referee 1: I'm not sure I understand why you discard the first 7 days as spin-up – perhaps because I'm used to prediction, where those 7 days are included in the forecast and would show large impacts of soil moisture initialization.

Author's response: Thank you very much. Spin-up is a concern when there is a lack of data or seasonal simulation (Rahman and Lu, 2015). Overestimating the spin-up period would lead to a loss of important information. Likewise, an underestimation will

lead to integrate errors in the analysis due to the fact that the model does not reach the dynamical equilibrium between the lateral forcing and the internal physical dynamic of the model. Yes, you're right, Anthes et al. (1989) demonstrated that regional models attain the dynamical equilibrium in 2-3 days spin-up period. However, Kang and al. (2014) by comparing different land surface schemes (BATS and CLM3) and different periods of spin-up to simulate June – July – August precipitations recommended 7 days as spin-up period. In this study, we used CLM4.5 as land surface scheme (Oleson et al., 2013) which has a more complex design. That's why we used 7 days as spin-up period.

3. Comments from referee 1: It would be prudent to discuss the implications of this work beyond a summary, perhaps in the concluding remarks.

Author's response: Thank you for your comment. Please be more specific in this comment to allow us to better understand the concern.

4. Comments from referee 1: You offer a comparison of CHIRPS and TRMM, and find large differences in the two datasets. How does this impact your results?

Author's response: Thank you for your comment. These differences the observation datasets have been revealed in several previous works over West Africa. For instance when comparing TRMM, GPCP and FEWS, Sylla et al. (2013) pointed out significant discrepancies between these products, whilst Nikulin et al. (2012) as well as Diallo et al. (2013) found large differences between gauge-based observations and satellite products. To minimize this impact of these discrepancies on our results we have chosen CHIRPS as a reference because of its high resolution.

5. Comments from referee 1: I have this problem a lot with manuscripts that include extreme indices - there area huge amount of indices to show, and this adds to length and can cause the reader to get lost in the paper as you go through each one. 21 figures is a lot! I like the way that you have isolated each index, but I think you could cut down on the detail slightly to save some words and not have the reader get lost in

the details.

Author's response: Thank you for your comment. We cut down on the detail in our analyzing in the revised manuscript.

Minor/Technical Comments:

1. Comments from referee 1: I noticed a number of grammatical and spelling errors in the manuscript, I suggest having someone read and edit the manuscript specifically for editorial remarks such as these. Author's response: Thank you for your comment. We did our best to improve the revised manuscript.

2. Comments from referee 1: You use a number of parenthetical references such as "impacts of the wet (dry) soil moisture on wet (dry) years etc." - I do not mind these at all, but sometimes the text is very difficult to read when they are used in excess. For example, line 464-468.

Author's response: Thank you for your comment. We reduce this style of writing in the revised version and make it easier to read.

3. Comments from referee 1: Define the lat and lon range of your domain(s).

Author's response: West Africa simulation domain Grid coordinates: 1: points=20748 (182x114) lon : -20 to 19.82 by 0.22 degrees_east lat: 0 to 24.86 by 0.22 degrees_north

Author's changes in manuscript: We did this following modification in the manuscript at Section 2.1 line 93 to 95: The integration of RegCM4 over the West African domain is shown in Fig. 1 with 18 vertical levels and 25 km (182x114 grid points; from 20°W-20°E and 5°S-21°N) of horizontal resolution.

4. Comments from referee 1: Line 102: Does this contradict your statement on line 23 of part 1? Perhaps rewording is necessary.

Author's response: Thank you for your comment. We rewrote the sentence.

Author's changes in manuscript: We did this following modification in the manuscript at Section 2.1 line 103: The sensitivity of initial soil moisture is not exceeded four months (Hong and Pan., 2000; Kim and Hong, 2006).

5. Comments from referee 1: Line 138 to 147: I believe you're talking about autocorrelation - neighboring grid points are spatially dependent. You do not necessarily need to resample, but you can estimate your n given autocorrelation - sometimes called effective sample size. I think you're using NCL in much of this manuscript (at least, your Figures look like NCL!), which has functions to calculate sample autocorrelation and equivalent sample size.

Author's response: Thank you for your comment. We do not seek to resample our data. We used the student t test to investigate the statistically significant differences between the control and the wet/dry sensitivity experiments at each grid cell as did by Liu and al (2014) in similar work over Asia. Due to the multiplicity problem of independent tests and the spatial dependency of neighboring grid points, the significant results can only be seen as a crude estimate. To justify this, Jager and Senviratne said that more reliable estimates of significance could be obtained using resampling methods proposed by Wilks (1997) for auto-correlated Fields. However, this is not feasible in our case due to the computational constraints associated with the size of our domain studied.

Author's changes in manuscript: We rewrote it to make it more comprehensive. We did this following modification in the manuscript at Section 2.1 line 140 to 146: The statistically significant differences has been tested between the control and the sensitivity experiments, we perform the two-tailed of the student's t-distribution at every grid points as did by Liu et al. (2014) in a similar work over Asia. Due to the multiplicity problem of independent tests and the spatial dependency of neighboring grid points, the significant results can only be seen as a crude estimate. Therefore, we perform the land point's area-weighted fraction with statistical significance of 10% level and we display the seasonally extreme indices maps during the years 2003 and 2004.
6. Comments from referee 1: Line 198: "Indicating that the number of wet days occurrence are occurred more likely not only in wet experiments but also in the dry experiments." I do not understand this sentence.

Author's response: Thank you for your comment. We would like to say that the number of wet days occurrence occurred not only in wet experiments but also in the dry experiments. Author's changes in manuscript: We did this following modification in the manuscript at Section 3.1 line 198 to199: Indicating that the number of wet days occurrence occurred not only in wet experiments but also in the dry experiments.

7. Comments from referee 1: I noticed that sometimes your section summaries only include some of your results - is there a way to make these more comprehensive without adding to length?

Author's response: Thank you for your comment. We rewrote these section summaries to make them more comprehensive. Please see through the revised manuscript.

References

Diallo I, Sylla MB, Gaye AT, Camara M (2013a) Comparaison du climat et de la variabilité interannuelle de la pluie simulée au Sahel par les modèles climatiques régionaux. Sécheresse. doi:10.1684/sec.2013.0382

Nikulin G, Jones C, Samuelsson P, Giorgi F, Asrar G, Büchner M, Cerezo-Mota R, Christensen OB, Déqué M, Fernandez J, Hänsler A, van Meijgaard E, Sylla MB, Sushama L (2012) Precipitation climatology in an ensemble of CORDEX-Africa Regional Climate simulations. J Clim. doi:10.1175/JCLI-D-11-00375.1

Liu, D., G. Wang, R. Mei, Z. Yu, and M. Yu(2014), Impact of initial soil moisture anomalies on climate mean and extremes over Asia, J. Geophys. Res. Atmos., 119, 529 – 545, doi:10.1002/2013JD020890.

Sylla MB, Giogi F, Coppola E, Mariotti L (2013b) Uncertainties in daily rainfall over Africa: assessment of gridded observation products and evaluation of a regional cli-

mate model simulation. Int J Climatol. doi:10.1002/joc.3551

Wilks DS (1997) Resampling hypothesis tests for autocorrelated fields. J Climate 10:65–82

Interactive
comment

[Figure]

West Africa

**Fig. 1:** Changes in daily soil moisture for 5 years (2001 to 2005) and their climatological mean during JJAS over West African domain, from dry (ΔDC) and wet (ΔWC) experiments with respect to their corresponding control experiment.

**Fig. 1.**

---

## Author Comment (AC2) · 3 Sep 2020

Major/Specific Comments: 1. Comments from reviewer 2: Model evaluation: The authors need to either demonstrate that the model used can reproduce precipitation or temperature extremes in the study region or provide a citation demonstrating this otherwise this model may not be a good tool for this research question. It's important that the evaluation be of precipitation extremes rather than the means or seasonal cycle (as in Koné et al. 2018) since that is what the authors are focusing on. Author's response: Thank you for your comment. The RegCM model is one of the most widely used models by the scientific community to reproduce mean and extreme climate almost anywhere in the world. In this study we evaluated its performance in West Africa for extreme climate. The model performs well in West Africa as well as in Asia in the representation of mean and extreme climate. The choice of a complex land surface model CLM4.5 coupled with RegCM4 need to be evaluated since it is not done before in climate extreme study over Africa. As compared with a previous study done by over Asia, RegCM4 reproduce well thee precipitation and temperature extremes over Africa.

Minor/Technical Comments:

1. Comments from reviewer 2: Minor points: Statistical significance: Perhaps I misunderstood the methods, but it seems like statistical significance can't be evaluated using this model setup (which is okay) but it shouldn't be presented as if it can. Each point only has a control year and two models run right? Please explain this further, the methods section does not provide enough detail here. What is your null distribution and what is your test distribution at each point? Author's response: Thank you for your comment. Our null hypothesis is the sample means are from the same population (i.e. H0: ave1=ave2). We used the student-t distribution. Rejection of the null hypothesis (i.e. acceptance of the alternative hypothesis) indicates that the sample means are from two different populations. Author's changes in manuscript: We did this following modification in the manuscript at Section 2.2 line 140 to146: we perform the two-tailed of the student's t-distribution at every grid points as did by Liu et al. (2014) in a similar work over Asia.

2. Comments from reviewer 2: PDF figures: In my opinion the PDFs don't add information and should probably be removed from both manuscripts to save space. The PDFs duplicate the spatial maps of changes, which provide more information, and double the number of figures presented. Author's response: Thank you for your comment. The use of PDFs is important because it gives important information such as how many grid are impacted, their highest value of biases and the quantification of the impact of the soil moisture initial conditions on the contrast between the years .The mean biases can't give such information.

3. Comments from reviewer 2: Pattern correlations in Table 3: It's not clear exactly how to interpret the pattern correlations for temperature. A value of 0.99 for every single value seems to imply that either there's an error in the calculation or that the metric is not useful. Are the temperature datasets this closely aligned, and if so would it be more useful to assess pattern correlation of temperature anomalies rather than the absolute temperature Author's response: Thank you for your comment. The pattern correlation coefficient is most of statistical tools used in modeling to assess the large scale correlation between two different product. This high value of the coefficient PCC is not new, many study with RegCM4 reveal its good large-scale representation of the temperature variable more than 0.9 (Diallo and al. 2013; Diallo and al 2016, Koné and al. 2018).

4. Comments from reviewer 2: I assume that the labels for TRMM should be EIN here as well. Author's response: Thank you for you. We don't know at which line this confusion is done but we improved the quality of the figures in this revised version. Please confim in which Figure or line the confusion has been done.

References:

Diallo I, Sylla MB, Gaye AT, Camara M, 2013. Comparaison du climat et de la variabilité interannuelle de la pluie simulée au Sahel par les modèles climatiques régionaux. Sècheresse 24 : 96-106. doi : 10.1684/sec.2013.0382

Diallo I., Giorgi F., Deme A, Tall M., Mariotti L., Gaye A., Projected changes of summer monsoon extremes and hydroclimatic regimes over West Africa for the twenty‑first century. Clim Dyn DOI 10.1007/s00382-016-3052-4, (2016).

Koné B., Diedhiou A., N'datchoh E. T., Sylla M. B. , Giorgi F., Anquetin S., Bamba A., Diawara A., and Kobea A. T.: Sensitivity study of the regional climate model RegCM4 to different convective schemes over West Africa. Earth Syst. Dynam., 9, 1261–1278. https://doi.org/10.5194/esd-9-1261-2018, 2018.

---

## Author Response (AR1)

**Reply to the comments of editor for the manuscript hess-2020-113.**

**Comments from the Editor:**

Thank you for providing your response to reviewers' comments. Please go ahead and submit the revised manuscript along with responses to the comments. In your response please also include page #, line # to specify where in the manuscript the changes are made in response to a given comment. Thank you and I look forward to seeing the revised manuscript.

**Reply to the comment of the Editor:**

*Thank you very much for your comments. Please find below, answers to reviewers' comments and changes including the page number and lines to specify where in the revised version the changes are made.*

**Reply to the comments of the referee 1 for the manuscript hess-2020-113**

**Major/Specific Comments:**

 1. **Comments:** As noted above, my main concern stems from your year choices. While this results in 6 experiments for comparison, I am not convinced that the results are robust given only a 2 year sample size. Moreover, I'm curious how these years were chosen -are they extreme wet and dry years? How often do years such as these occur? How is "wet" and "dry" defined?

**Author's response:**

*Thank you for your comment. For this revised version, as recommended, we re-run the simulations over 5 years (2001 to 2005) during the months of June to September over our West African domain. We superimposed the 5 years and their average in order to analyze the influence of initial soil moisture condition (Fig1 below, added in the revision version of Part 1). The Fig.1 (from Part 1 article) shows that the weakest and strongest impact of the dry experiments is found for 2003 and 2004 respectively. For a wet year, the impact of drying out soil moisture is quickly erased. While for a dry year the impact of the drying of the soil is accentuated. This means that 2003 and 2004 are respectively the wettest and driest years in dry experiment. However, for the wet experiments, the weakest impact is found for 2004, and the strongest impact is found for the years 2001, 2002 and 2004. In a dry year, the impact of soil humidification is very quickly erased, while in a wet year the impact of soil humidification is accentuated. The wet experiments confirm the result*

*obtained in dry experiments, 2003 and 2004 are wettest and driest years respectively. To estimate the limits of the impact of internal soil moisture forcing on the new dynamical core non-hydrostatic of RegCM4, we have used the two extreme years 2003 and 2004 (resp. the wettest and the driest years) among the 5 years. It is in the same context, several previous studies chosen two extreme years for their sensitivity study of initial soil moisture condition on the models. Hong and al. (2000) used in their study only two years (3 months per year) to investigate the impact of initial soil moisture over the North of America (in the Great Plains) during the two summers, May-June-July (MJJ) 1988 (corresponding to a drought in the Great plains) and MJJ 1993 (correspond to a flooding event). Over Asia, Kim and Hong (2006) in their paper "Impact of Soil Moisture Anomalies on Summer Rainfall over East Asia: A Regional Climate Model Study" used two contrasted years 1997 (below normal precipitation year) and 1998 (above normal precipitation year).*

[Figure]

**Fig.1**: Changes in daily soil moisture for 5 years (2001 to 2005) and their climatological mean during JJAS over West African domain, from dry (ΔDC) and wet (ΔWC) experiments with respect to their corresponding control experiment.

***Author's changes in manuscript**: We did this following modification in the manuscript at*

*Section 2.1 line 107 to 115:*

*As mentioned in part I, we performed these sensitivity studies to the initial conditions of soil moisture over our West African domain for June-July-August-September (JJAS) from 2001 to 2005 with a focus on two contrasted years 2003 (above normal precipitation year) and 2004 (below normal precipitation year). The two years 2003 and 2004 (resp. the wettest and the driest years among the 5 years) have been selected in the aim to estimate the limits of the impact of internal soil moisture forcing on the new dynamical core non-hydrostatic of RegCM4. Several previous studies used two extreme years for their sensitivity study of initial soil moisture condition on the models (e.g Hong and al., 2000; Kim and Hong,2006).*

**2. Comments from referee 1:**

I'm not sure I understand why you discard the first 7 days as spin-up – perhaps because I'm used to prediction, where those 7 days are included in the forecast and would show large impacts of soil moisture initialization.

**Author's response**: *Thank you very much. Spin-up is a concern when there is a lack of data or seasonal simulation (Rahman and Lu, 2015). Overestimating the spin-up period would lead to a loss of important information. Likewise, an underestimation will lead to integrate errors in the analysis due to the fact that the model does not reach the dynamical equilibrium between the lateral forcing and the internal physical dynamic of the model. Yes, you're right, Anthes et al. (1989) demonstrated that regional models attain the dynamical equilibrium in 2-3 days spin-up period. However, Kang and al.(2014) by comparing different land surface schemes (BATS and CLM3) and different periods of spin-up to simulate June – July – August precipitations recommended 7 days as spin-up period. In this study, we used CLM4.5 as land surface scheme (Oleson et al., 2013) which has a more complex design. That's why we used 7 days as spin-up period.*

**Author's changes in manuscript**: *We did this following modification in the manuscript at* **Section 2.1 line 118 to 123**: *Kang and al. (2014) by comparing different land surface schemes (BATS and CLM3) and different periods of spin-up to simulate June – July – August precipitations recommended 7 days as spin-up period. In this study, we used CLM4.5 as land surface scheme (Oleson et al., 2013) which has a more complex design.* The

first 7 days (Kang et al., 2014) are excluded in the analysis as a spin-up period.

**3. Comments from referee 1**: It would be prudent to discuss the implications of this work beyond a summary, perhaps in the concluding remarks.

*Author's response: Thank you for your comment. The section summaries and the discussion in the conclusion section have been improved to show implications of this study.*

*Author's changes in manuscript: We did this following modification in the manuscript: Please see new section summaries and in **Section 4 (in the conclusion) lines 666-669 we add**: This study helps to understand the impact of the initial soil moisture conditions on extreme events of precipitation and temperature in terms of intensity and duration over West Africa. It is a contribution to the improvement of extreme events forecasts in West Africa in highlighting the crucial role of initial soil moisture.*

**4. Comments from referee 1**: You offer a comparison of CHIRPS and TRMM, and find large differences in the two datasets. How does this impact your results?

**Author's response**: *Thank you for your comment. These differences between the observation datasets have been revealed in several previous works over West Africa. For instance when comparing TRMM, GPCP and FEWS, Sylla et al. (2013) pointed out significant discrepancies between these products, whilst Nikulin et al. (2012) as well as Diallo et al. (2013) found large differences between gauge-based observations and satellite products. We have chosen CHIRPS because of its high resolution and mainly because this product has been widely assessed and compared with other datasets and considered as more appropriate for study of extremes events in West Africa by Bichet et al 2018a, b and Didi et al 2020.*

- Bichet, A., & Diedhiou, A. (2018a). West African Sahel has become wetter during the last 30 years, but dry spells are shorter and more frequent. *Climate Research*, *75*(2), 155-162.
- Bichet, A., & Diedhiou, A. (2018b). Less frequent and more intense rainfall along the coast of the Gulf of Guinea in West and Central Africa (1981 2014). *Climate Research*, *76*(3), 191-201.
- Didi Sacré Regis M , Mouhamed, L., Kouakou, K., Adeline, B., Arona, D., Koffi Claude A, K., ... & Issiaka, S. (2020). Using the CHIRPS Dataset to Investigate Historical Changes in Precipitation Extremes in West Africa. *Climate*, *8*(7), 84.

*Author's changes in manuscript: We did this following modification in the manuscript at*

*Section 3.1.2 line 248 to 251*: *This shows a quite discrepancy among the observation datasets over West African domain. We have chosen CHIRPS because of its high resolution and mainly because this product has been widely assessed and used for study of extremes events in West Africa by Bichet et al. (2018a, b) and Didi et al. (2020).*

**5. Comments from referee 1**: I have this problem a lot with manuscripts that include extreme indices – there area huge amount of indices to show, and this adds to length and can cause the reader to get lost in the paper as you go through each one. figures is a lot! I like the way that you have isolated each index, but I think you could cut down on the detail slightly to save some words and not have the reader get lost in the details.

**Author's response**: *Thank you for your comment. We tried in this revised version to ease the reading in removing details where necessary.*

**Minor/Technical Comments:**

**1. Comments from referee 1:**

I noticed a number of grammatical and spelling errors in the manuscript, I suggest having someone read and edit the manuscript specifically for editorial remarks such as these.

**Author's response**: *Thank you for your comment. We did our best to improve the revised manuscript.*

**2. Comments from referee 1:** You use a number of parenthetical references such as "impacts of the wet (dry) soil moisture on wet (dry) years etc." - I do not mind these at all, but sometimes the text is very difficult to read when they are used in excess. For example, line 464-468.

**Author's response**: *Thank you for your comment. We reduced this style of writing in the revised version to make it easier to read.*

**3. Comments from referee 1**: Define the lat and lon range of your domain(s).

*Author's response*: *West Africa simulation domain Grid coordinates: 1: points=20748 (182x114) lon : -20 to 19.82 by 0.22 degrees east lat: 0 to 24.86 by 0.22 degrees_north.*

*Author's changes in manuscript*: *We did this following modification in the manuscript at*

***Section 2.1 line 96 to 98****: The integration of RegCM4 over the West African domain is shown in Fig. 1 with 18 vertical levels and 25 km (182x114 grid points; from 20◦W-20◦E and 5◦S-21◦N) of horizontal resolution.*

**4. Comments from referee 1**: Line 102: Does this contradict your statement on line 23 of part 1? Perhaps rewording is necessary.

**Author's response**: *Thank you for your comment. We rewrote the sentence. Instead of a "season", we specify in this revised version the number of months ("four months").*

**Author's changes in manuscript**: *We did this following modification in the manuscript at* **Section 2.1 line 106**: *The sensitivity of the initial soil moisture does not exceed four months (Hong and Pan., 2000; Kim and Hong, 2006).*

**5. Comments from referee 1**: Line 138 to 147: I believe you're talking about autocorrelation - neighboring grid points are spatially dependent. You do not necessarily need to resample, but you can estimate your n given autocorrelation - sometimes called effective sample size. I think you're using NCL in much of this manuscript (at least, your Figures look like NCL!), which has functions to calculate sample autocorrelation and equivalent sample size.

**Author's response**: *Thank you for your comment. We do not seek to resample our data. We used the student t test to investigate the statistically significant differences between the control and the wet/dry sensitivity experiments at each grid cell as did by Liu et al (2014) in similar work over Asia. Due to the multiplicity problem of independent tests and the spatial dependency of neighboring grid points, the significant results can only be seen as a crude estimate. To justify this, Jager and Senviratne said that more reliable estimates of significance could be obtained using resampling methods proposed by Wilks (1997) for auto-correlated Fields. However, this is not feasible in our case due to the computational constraints associated with the size of our domain studied.*

**Author's changes in manuscript**: We rewrote it to make it more comprehensive. We did this following modification in the manuscript at **Section 2.1 line 154 to 158**: The statistically significant differences has been tested between the control and the sensitivity experiments, we perform the two-tailed of the student's t-distribution at every grid points as did by Liu et al. (2014) in a similar work over Asia. Due to the multiplicity problem of

independent tests and the spatial dependency of neighboring grid points, the significant results can only be seen as a crude estimate.

**6. Comments from referee 1:** Line 198: "Indicating that the number of wet days occurrence are occurred more likely not only in wet experiments but also in the dry experiments." I do not understand this sentence.

**Author's response:** *Thank you for your comment. We would like to say that the increase of the number of wet days occurred not only in wet experiments but also in the dry experiments. As suggested above, we removed details to ease the reading.*

**Author's changes in manuscript**: *We did this following modification in the manuscript at* **Section 3.1.1 line 211 to 212**: *However, over the Guinea coast sub-region, both wet and dry experiments show a significant increase of R1mm, although weaker in the dry experiments.*

**7. Comments from referee 1**: I noticed that sometimes your section summaries only include some of your results - is there a way to make these more comprehensive without adding to length?

**Author's response**: *Thank you for your comment. We rewrote these section summaries to make them more comprehensive. Please see section summaries in the revised manuscript.*

**Reply to the comments of the referee 2 for the manuscript hess-2020-113**

**Major/Specific Comments:**

**1. Comments from reviewer 2**:

Model evaluation: The authors need to either demonstrate that the model used can reproduce precipitation or temperature extremes in the study region or provide a citation demonstrating this otherwise this model may not be a good tool for this research question. It's important that the evaluation be of precipitation extremes rather than the means or seasonal cycle (as in Koné et al. 2018) since that is what the authors are focusing on.

**Author's response**: *Thank you for your comment. The RegCM model is one of the most widely used models by the scientific community to reproduce mean and extreme climate a most anywhere in the world. In this study we evaluated its performance in West Africa for*

*extreme climate. The model performs well in West Africa as well as in Asia in the representation of mean and extreme climate. The choice of a complex land surface model CLM4.5 coupled with RegCM4 need to be evaluated since it is not done before in climate extreme study over Africa. As compared with a previous study done by over Asia, RegCM4 reproduce well thee precipitation and temperature extremes over Africa.*

**Minor/Technical Comments**:

**1. Comments from reviewer 2**:

Minor points: Statistical significance: Perhaps I misunderstood the methods, but it seems like statistical significance can't be evaluated using this model setup (which is okay) but it shouldn't be presented as if it can. Each point only has a control year and two models run right? Please explain this further, the methods section does not provide enough detail here. What is your null distribution and what is your test distribution at each point?

**Author's response**: *Thank you for your comment. Our null hypothesis is the sample means are from the same population (i.e.H0: ave1=ave2). We used the student-t distribution. Rejection of the null hypothesis (i.e. acceptance of the alternative hypothesis) indicates that the sample means are from two different populations.*

**Author's changes in manuscript**: *We did this following modification in the manuscript at **Section 2.2 line 154 to156**: The statistically significant differences has been tested between the control and the sensitivity experiments, we perform the two-tailed of the student's t-distribution at every grid points as did by Liu et al. (2014) in a similar work over Asia.*

**2. Comments from reviewer 2:** PDF figures: In my opinion the PDFs don't add information and should probably be removed from both manuscripts to save space.

The PDFs duplicate the spatial maps of changes, which provide more information, and double the number of figures presented.

**Author's response**: *Thank you for your comment. The use of PDFs is important because it gives important information such as how many grid points are impacted, what are their highest value of biases and help to quantify the impact of the initial soil moisture conditions. The mean biases can't give such information.*

**Author's changes in manuscript**: *We did this following modification in the manuscript at **Section 2.2 line 149 to 153**: To quantify the impact of soil moisture anomalies on climate extremes Liu et al. (2014) in their work over Asia, used the mean biases in 5 subregions,*

*while in our study we used the mean biases and the probability density function (PDF, Gao et al. (2016); Jaeger and Seneviratne (2011)) for this purpose to better capture how many grid points are impacted by initial soil moisture and their highest value.*

**2. Comments from reviewer 2:**

Pattern correlations in Table 3: It's not clear exactly how to interpret the pattern correlations for temperature. A value of 0.99 for every single value seems to imply that either there's an error in the calculation or that the metric is not useful. Are the temperature datasets this closely aligned, and if so would it be more useful to assess pattern correlation of temperature anomalies rather than the absolute temperature

**Author's response**: *Thank you for your comment. The pattern correlation coefficient is a common statistical tool used in modeling to assess the large-scale correlation between two different products. High value of PCC reveals a good large-scale spatial representation of the pattern of temperature by RegCM model (Diallo and al. 2013; Diallo and al 2016, Koné and al. 2018).*

**3. Comments from reviewer 2:**

I assume that the labels for TRMM should be EIN here as well.

**Author's response**: *Thank you for you. We don't know at which line this confusion is done but we improved the quality of the figures in this revised version.*

---

## Author Response (AR2)

**Reply to Editor on HESS-113 followed by answers to comments of Referee 1 and Referee 2**

**Editor Decision: Publish subject to revisions (further review by editor and referees)** (16 Nov 2020) by Shraddhanand Shukla
Comments to the Author:
Dear Authors,

Similar to the part 1, there are still some outstanding and valid comments from the reviewers that need to be addressed before this manuscript can be formally considered for publication. Please make sure that you provide detailed response to reviewers comments and revise the manuscript accordingly. I'll try to see the reviews again as per reviewers' availability.

Thanks again,
Shrad

**Author response to the editor comments**

*Dear Editor*

*Thank you for your comments and advice, as well as the reviewers who contributed with their feedback to improve the manuscript.*

*As suggested, we have sent the manuscript for a deep English Language editing (please see the certificate at the end of this document) to ease the reading and to avoid confusion due to language issue.*

*You will find in the next, our answers to the reviewers comments.*

*Thank you again and best wishes*

*Arona*

**CERTIFICATE OF ENGLISH EDITING**

This document certifies that the paper listed below has been edited to ensure that the language is clear and free of errors. The edit was performed by professional editors at Editage, a division of Cactus Communications, in cooperation with Taylor & Francis Group. The intent of the author's message was not altered in any way during the editing process. The quality of the edit has been guaranteed, with the assumption that our suggested changes have been accepted and have not been further altered without the knowledge of our editors.

**Title**

Influence of initial soil moisture in a Regional Climate Model study over West Africa. Part 2: Impact on the climate extremes

**Authors**

Brahima Koné, Arona Diedhiou, Adama Diawara, Sandrine Anquetin, N'datchoh Evelyne Touré, Adama Bamba, and Arsene Toka Kobea

**Order No.**

RODIE_2

**EDITINGSERVICES**
Supporting Taylor & Francis authors

[Figure]

Signature

*Vikas Narang*

Vikas Narang,
Chief Operating Officer,
Editage

Date of Issue
**November 30, 2020**

**Taylor & Francis Editing Services**

www.tandfeditingservices.com
support@tandfeditingservices.com

**Reply to the comments of referee 1 on HESS-113**

This is the second revision on "Influence of initial soil moisture in a Regional Climate Model study over West Africa. Part 2: Impact on the climate extremes." I found the manuscript to be improved from the first, and the authors provided context on the review comments noting their experiment design. Overall, I think the results will be helpful for the field. I think the manuscript still suffers from some organizational issues, and if the authors can improve the organization the manuscript will be greatly improved.
I recommend minor revisions mostly for organization and clarity. While I only have a few comments, I think comment 1 will take some time and care to respond to.

**Major/Specific Comments:**
   1. **Comments:**
This manuscript is attempting to show a large amount of comparisons. For instance, the authors use 2 observational datasets that are compared, the RegCM4 control run compared to observations, stratify the analysis based on the two extreme years, and provide results based on their idealized soil moisture experiments. I appreciate the authors summarizing the results from each of their indices because it helped a lot with clarity, but I think the authors could go a step further and make it more clear what the main goal and results of this study are. i.e., Are the authors trying to mainly show the ability of RegCM4 to represent the extreme indices, the differences between two extreme years, or the idealized experiments? It could be all of these, but if that's the case I think your main points get somewhat lost in the details. In addition, while I again appreciate the authors care in comparing two observational datasets, I think it adds unnecessary information to the manuscript, and it also doesn't make sense when the authors say things like "TRMM is biased compared to CHIRPS" because they have not provided any reasoning why one dataset is preferred over the other. To sum up this comment, I think the manuscript would benefit from the authors picking a clear goal.

*Author's response: Thank you for your comment.*

*First, we have sent the manuscript for a deep English Language editing (please see the certificate at the end of this document) to ease the reading and to avoid confusion due to language issue.*

*Then, the introduction has been improved and the objective of this paper was made clearer as follows (lines 60-74):* In part 1, the influence of initial soil moisture on the climate mean was based on a performance assessment of the Regional Climate Model coupled with the complex Community Land Model (RegCM4-CLM4.5) performed by Koné et al. (2018), where the ability of the model to reproduce the climate mean has been validated. In Part 2, before starting to study the influence of initial soil moisture on climate extremes, it was necessary to assess the performance of RegCM4-CLM4.5 in simulating the ten (10) indices of temperature and precipitation extremes used in this study. This has never been done before in West Africa with this version of RegCM with a non-hydrostatical scheme; therefore, we separated the work in

two parts, a first one assessing the ability of the model to simulate the climate extreme indices, and a second one investigating how and what is the time limit of the effect of initial soil moisture condition on the magnitude or duration of these climate extremes. The manuscript is organized as follows: Section 2 describes the RegCM4 model, the experimental design, and the methodology used in this study; Section 3 presents results of the two parts of the work and Section 4 documents the main conclusions.

*Finally, we added in this revised manuscript the reason of the choice of CHIRPS product for precipitation and NOAA-CPC daily temperature from the GTS as reference data used to validate the model. We removed some details in this revised manuscript.*

**Author's changes in manuscript**: *We did this following modification in the manuscript*

> *At the section 3.1.1 line 183 to 185:* We have chosen CHIRPS as reference in this study, because of its high resolution and mainly because this product has been widely assessed and compared with other datasets and considered as more appropriate for study of extremes events in West Africa by Bichet et al. (2018a, b) and Didi et al. (2020).

> *In the section 3.2.1 line 446 to 451:* Fan Y. and Huug van den Dool (2008) in their work showed that the Reanalysis 2 m temperature data sets may not be suitable for model forcing and validation. We have chosen NOAA-CPC GTS observation dataset as reference in this study over ERA-Interim reanalysis, because NOAA-CPC GTS consists of a blending of satellite-based data collection and in situ data archive available in the GTS (Global Telecommunication System).

**2. Comments:**

I'm still having some issues/concerns with your significance tests. Almost the entire region is significantly different in most of your Figures, to the point where it would be easier to show areas where there is NO significant difference. You did use a 90% confidence level, perhaps this is too low given your datasets? Is your sample size too low to form a proper significance test?

**Author's response**: *Thank you for the comment.* We confirm that in these figures, *only values passing the 10% significance test are dotted.*

**Minor Comments:**

1. ***Comment:*** As in your first manuscript, I strongly suggest very careful editing. I gave a large amount of examples in my first review, and I'm going to give 1 example here: (Line 154) The statistically significant differences has been tested between the control and the sensitivity experiments, we perform the two-tailed of the student's t-distribution at every grid points as did by Liu et al. (2014) in a similar work over Asia. Can be rewritten to something similar to the following: Significance of differences was tested for the control vs. sensitivity experiments. We used a two-tailed Student's t-test at each grid point as in Liu et al. (2014).

   ***Author's response****: Thank you for the comment and suggestion which contributed to improve the manuscript. As told above, we have sent the manuscript for a deep English Language editing (please see the certificate at the end of this document). Your suggestion has been taken into account. Please see lines 154-155:* Significance of differences was tested for the control vs. sensitivity experiments. We used a two-tailed Student's t-test at each grid point as in Liu et al. (2014) over Asia.

2. ***Comment:*** *Table 2: This is a lot of data to look at. Is there a way to highlight some of the values you want to draw the reader to? Such as using an asterisk, bolding, italics, etc.*

   ***Author's response****: Thank you for the suggestion. We highlighted the values that we want to draw the reader's attention.*
   **Author's changes in manuscript**: *See Table 2.*

3. ***Comment:*** Table 3: Same comment as above.
   ***Author's response****: Thank you for your advice. We have highlighted the values to which we would like to draw the reader's attention.*
   **Author's changes in manuscript**: *See Table 3.*

**References:**

Bichet, A., & Diedhiou, A. (2018a). West African Sahel has become wetter during the last 30 years, but dry spells are shorter and more frequent. *Climate Research*, *75*(2), 155-162.

Bichet, A., & Diedhiou, A. (2018b). Less frequent and more intense rainfall along the coast of the Gulf of Guinea in West and Central Africa (1981 2014). *Climate Research*, *76*(3), 191-201.

Didi Sacré Regis M , Mouhamed, L., Kouakou, K., Adeline, B., Arona, D., Koffi Claude A, K., ... & Issiaka, S. (2020). Using the CHIRPS Dataset to Investigate Historical Changes in Precipitation Extremes in West Africa. *Climate*, *8*(7), 84.

Fan Y., and van den Dool H. : A global monthly land surface air temperature analysis for 1948 - present, J. Geophys. Res. 113, D01103, doi: 10.1029/2007JD008470, 2008.

**CERTIFICATE OF ENGLISH EDITING**

This document certifies that the paper listed below has been edited to ensure that the language is clear and free of errors. The edit was performed by professional editors at Editage, a division of Cactus Communications, in cooperation with Taylor & Francis Group. The intent of the author's message was not altered in any way during the editing process. The quality of the edit has been guaranteed, with the assumption that our suggested changes have been accepted and have not been further altered without the knowledge of our editors.

**Title**

Influence of initial soil moisture in a Regional Climate Model study over West Africa. Part 2: Impact on the climate extremes

**Authors**

Brahima Koné, Arona Diedhiou, Adama Diawara, Sandrine Anquetin, N'datchoh Evelyne Touré, Adama Bamba, and Arsene Toka Kobea

**Order No.**

RODIE_2

**EDITINGSERVICES**
Supporting Taylor & Francis authors

Signature

*Vikas Narang*

Vikas Narang,
Chief Operating Officer,
Editage

Date of Issue
**November 30, 2020**

[Figure]

**Taylor & Francis Editing Services**

www.tandfeditingservices.com
support@tandfeditingservices.com

**Reply to the comments of referee 2 on HESS-113**

*The authors have not addressed my previous concerns. See below for portions of my original comment, followed by the author response, and my follow-up comment. I still have 4 concerns with the manuscript.*

**1. Original comment:** *"Model evaluation: The authors need to either demonstrate that the model used can reproduce precipitation or temperature extremes in the study region or provide a citation demonstrating this otherwise this model may not be a good tool for this research question. It's important that the evaluation be of precipitation extremes rather than the means or seasonal cycle (as in Koné et al. 2018) since that is what the authors are focusing on.*

> **Author's response:** *Thank you for your comment. The RegCM model is one of the most widely used models by the scientific community to reproduce the mean and extreme climate around the world. In this study, we evaluated its performance in West Africa for extreme climate. The model performs well in West Africa as well as in Asia for the representation of mean and extreme climate".*

**Follow up comment:** *Please provide a reference for this, particularly with respect to climate extremes. Replying to my comment by stating it in an author response without reference is inadequate and unsupported. I don't mean to be stubborn, but the authors' response is dismissive.*

> **Author's response**: *Thank you for your comment. Our apologies for the inconvenience this response may have caused, this was not really intentional. The model performs well in West Africa as well as in Asia in the representation of mean and extreme climate (Thanh and al. (2016); Gu H. et al. (2020); Liu and al. (2014); Diba et al. (2019)*

**2. Statistical significance:** *The procedure for calculating statistical significance remains unclear. Do the authors pool all points in a given year into a single distribution to test significance of the test year against the reference? If so, this makes sense, but it does not make sense how they stipple point-wise statistical significance as in the bottom panels of Figure 2. With point-wise statistical significance the authors are apparently comparing the three model ensemble observations at each point against a single reference value? It's not possible to do statistical significance using 3 observations. Using a student-t distribution with three values is not appropriate. Please clarify, how many observations are in the reference distribution at each point in space and how many observations are in the wet or dry experimental ensemble at each point in space. As I mentioned in my original comment, it's okay if the authors simply remove the statistical significance.*

> **Author's response**: *Thank you for your comment. The statistically significant differences has been tested between 2 variables the sensitivity experiments (wet or dry) and the control (reference). Assume*

X (wet) and Y (CTRL).  The null hypothesis that the sample temporal means are from the same population (i.e. H0: aveX=aveY).
The procedure to calculate the significance is:
-compute the temporal means at each grid point.
- Specify a critical significance level for the student's t-distribution and test if the means are different at each grid point.
The dotted area shows changes with statistical significance of a given level.

**3. Pattern correlation in Table 3:** *I understand that pattern correlation is a common statistical tool, but I don't believe it's appropriate to use the pattern correlation of absolute temperature values. It would be more appropriate to conduct a pattern correlation of anomalies of each variable rather than the absolute value.*

**Author's response**: *Thank you for your comment. In climate modeling we often used to compute the correlation pattern between two samples (For example between model outputs and observation datasets). This spatial correlation (PCC) is computed with respect to a reference. We don't know if this is what you call computing correlation of anomalies of each variable.*

**4. My final comment was for Table 3.** *I believe the left columns in Table 3 should not read "TRMM_2003" but rather "EIN_2003" correct? This table is evaluating the daily temperature anomalies from the EIN reanalysis, not the TRMM observations*

**Author's response**: *Yes, you are right. Thank you very much. We corrected it. Please see the Table 3.*

**References:**

*Liu, D., G. Wang, R. Mei, Z. Yu, and M. Yu(2014), Impact of initial soil moisture anomalies on climate mean and extremes over Asia, J. Geophys. Res. Atmos., 119, 529 – 545, doi:10.1002/2013JD020890.*

*Thanh N.-D.,  Fredolin T. T., Jerasorn S., Faye C., Long T.-T., Thanh N.-X., Tan P.-V., Liew J., Gemma N., Patama S., Dodo G. and Edvin A.: Performance evaluation of RegCM4 in simulating extreme rainfall and temperature indices over the CORDEX-Southeast Asia region. Int. J. Climatol. 37: 1634–1647. Published online 28 June 2016 in Wiley Online Library (wileyonlinelibrary.com) DOI: 10.1002/joc.4803, 2017.*

*Gu, H.; Wang, X. Performance of the RegCM4.6 for High-Resolution Climate and Extreme Simulations over Tibetan Plateau. Atmosphere 2020, 11, 1104.*

*Diba, I., Camara, M. and Diedhiou, A. (2019) Impacts of the Sahel-Sahara interface reforestation on West African climate: intra-annual variability and extreme temperature*

*events. Atmospheric and Climate Sciences, 9, 35 – 61.*
*https://doi.org/10.4236/acs.2019.91003.*

*Diba, I., Camara, M. and Sarr, A.B. (2016) Impacts of the Sahel-Sahara interface reforestation on West African climate: intraseasonal variability and extreme precipitation events. Advances in Meteorology, 2016, 3262451. http://dx.doi.org/10.1155/2016/3262451.*

*Saley, I.A.; Salack, S.; Sanda, I.S.; Mounkaila, S.M.; Bonkaney, A.L.; Ly, M.; Fodé, M. The possible role of the Sahel Greenbelt on the occurrence of climate extremes over the West African Sahel. Atmos. Sci. Lett. 2019, 20,e927.*

**CERTIFICATE OF ENGLISH EDITING**

This document certifies that the paper listed below has been edited to ensure that the language is clear and free of errors. The edit was performed by professional editors at Editage, a division of Cactus Communications, in cooperation with Taylor & Francis Group. The intent of the author's message was not altered in any way during the editing process. The quality of the edit has been guaranteed, with the assumption that our suggested changes have been accepted and have not been further altered without the knowledge of our editors.

**Title**

Influence of initial soil moisture in a Regional Climate Model study over West Africa.
Part 2: Impact on the climate extremes

**Authors**

Brahima Koné, Arona Diedhiou, Adama Diawara, Sandrine Anquetin, N'datchoh
Evelyne Touré, Adama Bamba, and Arsene Toka Kobea

**Order No.**

RODIE_2

**EDITING**SERVICES
Supporting Taylor & Francis authors

Signature

*Vikas Narang*

Vikas Narang,
Chief Operating Officer,
Editage

Date of Issue
**November 30, 2020**

[Figure]

**Taylor & Francis Editing Services**

www.tandfeditingservices.com
support@tandfeditingservices.com

---

## Author Response (AR3)

**Reply to the comments of referee 1 for Second Revision of HESS-113**

Thank you for revising the manuscript. As in the part 1 of this study, there are still some remaining comments that need to be addressed. Please see the comments of reviewer #2, especially the comment related to significance test. Please make sure to provide response to each of their comment and specify what changes were made or not, in the manuscript.

**Major/Specific Comments:**

This is the third revision of HESS 113. Overall I find Part-II to be interesting, but I don't feel that the authors fully responded to my comments in my second revision. Also, as noted in my review of HESS 112, while I appreciate the authors sending the manuscript for editorial review, I think the authors still need to read over the paper for some missed edits, some of which are noted below. Overall my recommendation is minor revisions as again I think these are mostly cosmetic suggestions, but please note that the authors should carefully respond to each comment.

1. *Comments:*

In my second review, comment 2, I stated: I'm still having some issues/concerns with your significance tests. Almost the entire region is significantly different in most of your Figures, to the point where it would be easier to show areas where there is NO significant difference. You did use a 90% confidence level, perhaps this is too low given your datasets? Is your sample size too low to form a proper significance test?
The authors response did not fully answer the questions posed. One could simply remove the significance test at this point, but if not, then do the authors feel that 90% CL is appropriate here, and why? In addition, I feel like your sample size is far too low to perform a proper significance test, but I'm unsure of the sample size used as I do not see it stated.

*Thank you very much. As agreed in Part 1 of this study, the main shortcoming was that we performed the significance test with monthly values leading to samples of small size. In this revised version, instead of doing the Student t-test with monthly means, we did it with daily values (from June to September) for each year (2003 and 2004) and thus, with samples of 115 days (without the 7 days spin-up period).*

*The two paragraphs below were added (lines 127-141) at the end of the section 2.1 (Model description and numerical experiments) to introduce the experiments design and the Student t-test as follows:*

**In the part 1 of this study, we designed three experiments (reference, wet, and dry), each with a set of five (5) simulations starting from June 1st to September 30th. The difference between these three experiments is the change in the initial soil moisture condition (reference initial soil moisture condition, wet initial soil moisture condition, and dry initial soil moisture condition) during the first day of the simulation (June 1st, 2001, 2002, 2003, 2004, and 2005) over the West African domain. Then, we selected the two years most affected by the wet and dry initial soil moisture conditions (2003 and 2004) to estimate the limits of the impact of the internal soil moisture forcing on the new non-hydrostatic dynamic core of RegCM4.**

*For these two years most sensitive to soil moisture initial conditions, the Student t-test is used to compare the significance of changes in climate extreme indices between a wet or dry sensitivity test (sample 1) and the control (sample 2) in assuming that this method performs well for climate simulations (Damien et al., 2014) and knowing that it is extensively used for climatological analysis (Menedez et al., 2019; Talahashi and Polcher, 2019). In this study, the t-test at the 95% confidence level was used to consider statistically significant.*

2. **Comments:**

There is an exclamation point on line 156 after the word "neighboring"

*Thank you. In this revised version, this sentence has been deleted. A new paragraph on the Student t-test has been added at the end of the section 2.1, as mentioned above (in the answer to the first comment).*

3. **Comments:**

Line 191: "The TRMM datasets underestimate..." Compared to CHIRPS?

*Thank you for the comment. In agreement with your comment 5 (below), this sentence has been deleted in this revised version. We agree with your suggestion and remove the comparison between CHIRPS and TRMM to ease the reading and to be more focus on the aim of the study.*

4. **Comments:**

Line 193 should read "The strongest underestimation was found over the central Sahel..."

*This sentence referred to the comparison between the two precipitation products. As said above, it has been deleted in this revised version.*

5. **Comments:**

Given the authors response to my note in my second review, comment 1, on the use of the two observational datasets TRMM and CHIRPS, I'm unsure why this comparison is indeed necessary for this particular manuscript. If in fact CHIRPS is considered more appropriate for extremes in West Africa with the given references, why re-assess it here? It may clarify things to remove this comparison, unless the authors think it is strictly necessary to include. If the authors do think that this comparison is necessary to include, then very clear reasoning should be provided.

*Thank you for the comment. We agree with your suggestion and we removed the comparison between CHIRPS and TRMM to ease the reading and to be more focus on the aim of the study. In this revised version, we used CHIRPS products and we introduced them as follows (Lines 147-149):* **We have chosen CHIRPS as reference in this study, mainly because this product has been widely assessed and used for the study of extreme events in West Africa by Bichet et al. (2018a, b) and Didi et al. (2020).**

6. **Comments:**

Are there similar references that discuss the ability of the temperature datasets over West Africa? I could see this as a reason for including the comparison if there are no other references, but again this does not seem to be the focus of this particular manuscript.

*Thank you for the comment. You're right. In this revised version, we removed the comparison between the temperature datasets to be more focus on the aim of the study. While the temperature product used remains the same, we changed its name in this version (CPC-T2m instead of common name GTS temperature dataset) to be more specific. The following lines have been added to introduce the T2m product (Lines 150-157):* **We validated the 2-m temperature using the NOAA/NCEP/CPC daily maximum and minimum global surface air temperature. The NOAA/NCEP/CPC global daily surface 2-m air temperature (CPC-T2m) is a land-only gridded global daily maximum (Tmax) and minimum (Tmin) temperature analysis from 1979 to the present, available at two spatial of 10 min × 10 min and 0.5° × 0.5° (latitude × longitude). This product provides an observational T2m estimate for climate monitoring, model evaluation, and forecast verification (Fan Y. and Huug van den Dool, 2008; Pan et al., 2019). In this study, the daily Tmax and Tmin are used at spatial resolution 0.5° × 0.5°..**

7. ***Comments:***

As in Part-I, I suggest again noting that these experiments are done in a highly-idealized framework and are intended to show the potential impact of very strong soil moisture conditions on extremes, and should thus be used as a guide or first look at the influence of soil moisture on extremes.

*Thank you very much. The following paragraph has been added at the end of the conclusions section (lines 566-572) as follows:*

**This study is the first to investigate the impact of soil moisture initial conditions on climate extreme indices over West Africa. These experiments were done in a highly-idealized framework and were intended to show the potential impact of very strong soil moisture initial conditions on climate extremes. Consequently, it should be considered as a first overview of the influence of initial soil moisture on climate extremes with a RCM (RegCM4). In perspectives, this study will benefit from being performed in a multi-model framework with several RCMs within CORDEX-Africa initiative (Coordinated Regional Downscaling Experiment).**

---

## Author Response (AR4)

**Reply to the comments of referee 1 for Second Revision of HESS-113**

The authors have done a great job with edits and responding to my comments. I found the manuscript much improved from the previous iteration and much easier to follow. My final comments are similar to those from HESS-112 and I suggest minor revisions at this stage.

**Minor Comments:**

1. *Comments:*

Line 511, I suggest deleting "In perspectives" from this sentence.

> *Author's response*: Thank you. We removed "In perspectives" from this sentence in the revised manuscript.

> *Author's changes in manuscript*:

> Lines 511 to 513: *This study will benefit from being performed in a multi-model framework with several RCMs within CORDEX-Africa initiative (Coordinated Regional Downscaling Experiment).*

2. *Comments:*

As in HESS-112, it may be worthwhile to add a brief sentence on how your ensemble members are generated.

> *Author's response:* Thank you for your suggestion. We added a sentence on how the ensemble members are generated in the revised manuscript.

> *Author's changes in manuscript*:

> Please see lines 109 to 114: "We designed three experiments (reference, wet, and dry), each with an ensemble of five (5) simulations (2001, 2002, 2003, 2004, and 2005) starting from June 1st to September 30th. The difference between these three experiments is the change in the initial soil moisture condition (reference initial soil moisture condition, wet initial soil moisture condition, and dry initial soil moisture condition) during the first day of the simulation over the West African domain."